# Phenotyping single-cell motility in microfluidic confinement

Samuel A Bentley[1,2,3†], Hannah Laeverenz-Schlogelhofer[1,2†],
Vasileios Anagnostidis[1,3,4], Jan Cammann[5], Marco G Mazza[5,6], Fabrice Gielen[1,4*],
Kirsty Y Wan[1,2*]

[1]Living Systems Institute, University of Exeter, Exeter, United Kingdom; [2]Mathematics and Statistics, University of Exeter, Exeter, United Kingdom; [3]Biosciences, University of Exeter, Exeter, United Kingdom; [4]Physics and Astronomy, University of Exeter, Exeter, United Kingdom; [5]Interdisciplinary Centre for Mathematical Modelling and Department of Mathematical Sciences, Loughborough University, Loughborough, United Kingdom; [6]Max Planck Institute for Dynamics and Self-Organization (MPIDS), Göttingen, Germany

*For correspondence:
F.Gielen@exeter.ac.uk (FG);
K.Y.Wan2@exeter.ac.uk (KYW)

†These authors contributed equally to this work

Competing interest: The authors declare that no competing interests exist.

**Abstract** The movement trajectories of organisms serve as dynamic read-outs of their behaviour and physiology. For microorganisms this can be difficult to resolve due to their small size and fast movement. Here, we devise a novel droplet microfluidics assay to encapsulate single micron-sized algae inside closed arenas, enabling ultralong high-speed tracking of the same cell. Comparing two model species - *Chlamydomonas reinhardtii* (freshwater, 2 cilia), and *Pyramimonas octopus* (marine, 8 cilia), we detail their highly-stereotyped yet contrasting swimming behaviours and environmental interactions. By measuring the rates and probabilities with which cells transition between a trio of motility states (smooth-forward swimming, quiescence, tumbling or excitable backward swimming), we reconstruct the control network that underlies this gait switching dynamics. A simplified model of cell-roaming in circular confinement reproduces the observed long-term behaviours and spatial fluxes, including novel boundary circulation behaviour. Finally, we establish an assay in which pairs of droplets are fused on demand, one containing a trapped cell with another containing a chemical that perturbs cellular excitability, to reveal how aneural microorganisms adapt their locomotor patterns in real-time.

## Editor's evaluation

This paper reports on the development of an impressive microfluidic platform for the study of motility, and motility transitions, exhibited by single algal cells in circular confinement. Building on previous work that showed a three-state motility repertoire for certain green algae, the present work uses extremely long time series and a variety of physical perturbations to show how those dynamics can be altered by environmental conditions. The work will be of interest to a wide range of scientists studying motility and non-equilibrium dynamics.

## Introduction

All lifeforms are environmentally intelligent; even single cells sense and respond to their environment (*Jennings, 1901*). How living systems interact with the external world is a fundamental question that has fascinated researchers for centuries. The exploratory behaviours of living beings can be extraordinarily complex and context-dependent, and yet also stereotyped. Digitally-assisted and non-invasive tracking methods have revolutionised the study of behaving animals (*Berman, 2018*; *Mathis et al.,*

*2018*). Algorithms for pose and gait-estimation based on computer vision and machine learning have bolstered the utility of animal models for neuroscience, providing a highly quantitative and objective framework for measuring behaviour. These nascent approaches are now being used in combination with genetic manipulation, to resolve how an organism's sensory architecture (neural pathways, nervous systems) may be coupled to motor actuators (e.g. muscles) to drive motor actions (*Brown et al., 2013*).

Surprisingly, nervous systems are not necessary for the emergence of complex search patterns or behavioural motifs (*Coyle et al., 2019*; *Wan, 2020*). Micoorganisms also adopt highly stereotyped locomotion strategies, often employing motile appendages: bacterial flagella for swimming, pili for twitching and swarming, and in many eukaryotes, cilia for self-propulsion. Microbial motility is strongly influenced by boundaries, interfaces, and other obstacles (*Ostapenko et al., 2018*; *Kantsler et al., 2013*). Microorganisms naturally encounter heterogeneous environments, such as soil, sea ice, or porous materials that constitute complex networks of confining tubes, spaces, or interstices. By replicating such microenvironments in the laboratory, realistic situations can be created for tracking and understanding the mechanisms of movement and behavioural adaptation (*Théry et al., 2021*; *Bhattacharjee and Datta, 2019*).

Despite their small size, microswimmers self-propel at speeds of several tens (even hundreds) of body lengths per second, posing a formidable challenge for live-cell microscopy and tracking. Typically, the same individual can only be observed for relatively short periods, and motility statistics are then coarse-grained across entire populations. Across different species, even across different individuals of the same clonal population, microbes exhibit significantly divergent behaviours. This single-cell heterogeneity may be hidden if only population averages are considered. However, long-term tracking of the same individual was not possible until very recently (*Hageman et al., 2019*; *Ostapenko et al., 2018*; *Codutti et al., 2022*).

What are the behavioural signatures that distinguish one microswimmer from another? To address this, we phenotype two species of motile algae, and decode their responses to diverse environmental cues. The first, *Chlamydomonas reinhardtii* (hereafter CR) is a freshwater biflagellate and a model organism for motility studies and cilia biology (*Sasso et al., 2018*). The second, *Pyramimonas octopus* (hereafter PO), is a marine octoflagellate, which has a unique tripartite run-stop-shock motility pattern (*Wan and Goldstein, 2018*). Harnessing modern microfluidics technology to access long-term behaviour (*Brückner et al., 2019*; *Tweedy et al., 2020*), we developed a platform to stably encapsulate algae inside droplets contained in shallow microwells. Here, cells can explore and interact with their local environment over much longer timescales than previously possible. Single-cell responses were evaluated against physical confinement, light, and chemical stimulation. For the latter, this is achieved by fabricating a tailored device to directly perturb the chemical environment of a trapped cell, by droplet pairing and fusion controlled by surfactant-induced interface destabilization (*Niu et al., 2009*; *Frenz et al., 2008*; *Fradet et al., 2011*).

From these long-time recordings, we then explore universal metrics of microbial motility, including speed, gait switching probabilities, and emergent spatial probability fluxes in the time-averaged motion of single cells. We show how these behaviours are effectively captured by a model of a roaming cell in a circular corral. Taken together, our approach integrates experimental and computational techniques to provide an exciting opportunity to understand how individual living cells interact with local environmental cues at the microscopic scale.

## Results
### Microfluidic trapping of single motile algae
We encapsulated CR or PO cells in droplet microwells of four different diameters (Ø = 40, 60, 120, 200 μm). To study the cells' baseline motility (in the absence of stimulation), we imaged in red light to prevent phototactic effects (long-pass IR filter). According to the absorption spectra of CR photoreceptors (ChR1 and ChR2), cells are insensitive to wavelengths of > 600 nm (*Harz and Hegemann, 1991*). The identity of the photoreceptor in PO has not yet been confirmed, but is likely to be similar to CR, since green algal eyespots are highly conserved (*Kreimer, 2009*).

Only wells containing precisely one cell were imaged (see Materials and methods). These water-in-oil droplets were stably trapped and retained their shape and volume for more than 1 hr. The

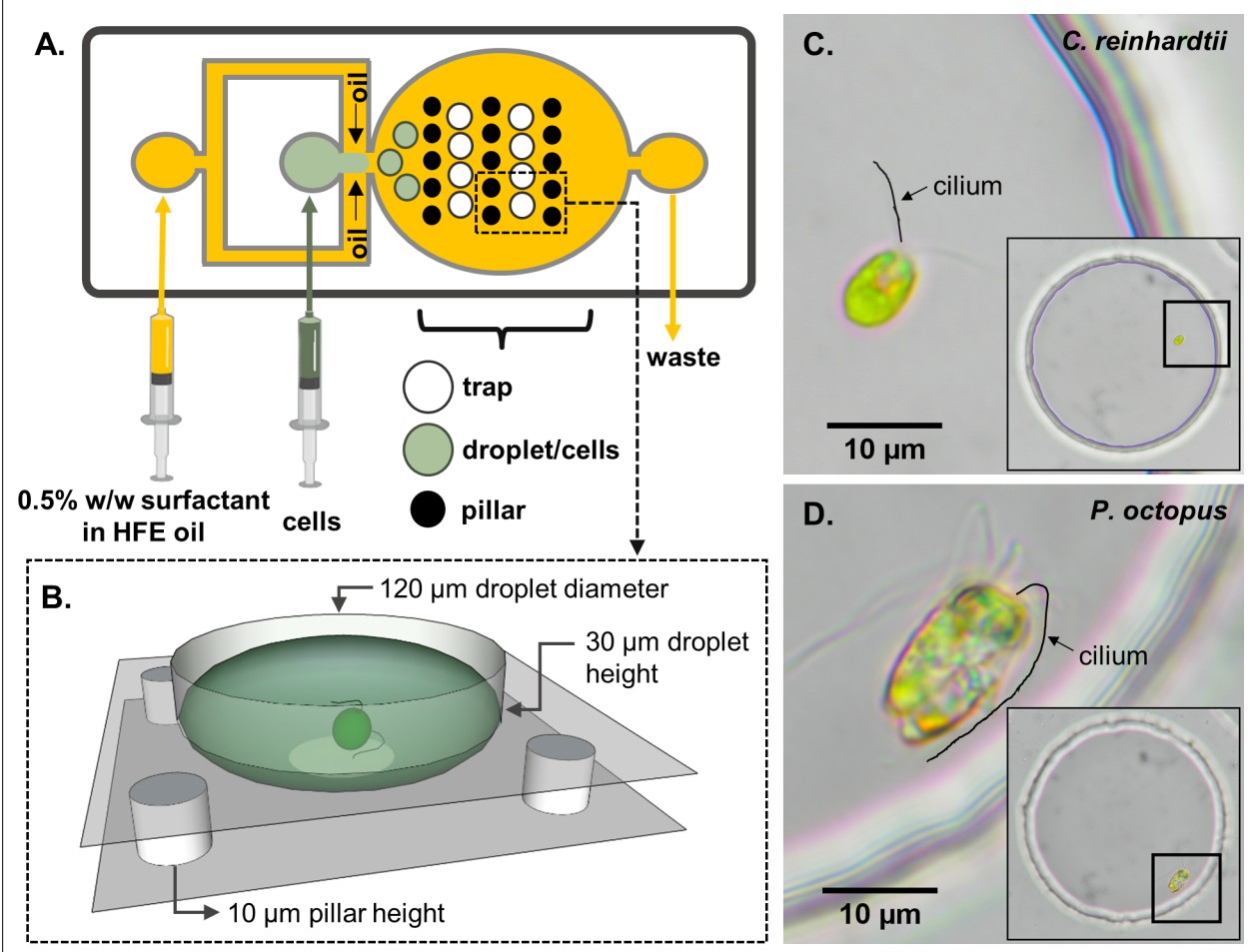

**Figure 1.** Schematic of the experimental set-up. (**A**) A two-layer microfluidic device with embedded single-cell traps, and syringes used for perfusion of the carrier oil phase and an aqueous suspension containing live motile cells. (**B**) 3D rendering of a single trap in which a cell can be stably trapped and imaged for hours. To demonstrate variability in swimming behaviour, we studied two species of motile algae, images show respectively: (**C**) a single *Chlamydomonas reinhardtii* (CR) cell, and (**D**) a single *Pyramimonas octopus* (PO) cell, in each case trapped within a 120 µm-diameter circular well. (Cilia positions are highlighted by manual tracings.).

trapping procedure was sufficiently gentle and did not damage the cilia or cause spontaneous deciliation. Compared to the cell-scale, the arenas are shallow (~ 30 µm deep in total) and quasi-2D, so that swimming motion is largely restricted to the plane (*Figure 1A and B*). We trapped each cell for a total of 1 hr, recording at 5-min intervals ($N = 5$ cells per trap size, 6 time points each) at 500 frames per second. High-speed imaging was necessary to resolve rapid motility changes. Cell-centroids were tracked with Trackmate in ImageJ and analysed with custom scripts (see Materials and methods).

The two species display very different swimming behaviours. Typical trajectories and speeds are displayed in *Figure 2*. Both species have front-mounted cilia (*Figure 1C and D*). CR executes a breast-stroke gait where the two cilia alternate between long-periods of in-phase synchrony, and phase slips when one cilium transiently undergoes altered beating (*Wan et al., 2014*). This leads to noisy trajectories with straight runs whenever the cilia are synchronised, and turns whenever synchrony is lost. Asynchronous cilia dynamics are associated with reduced swimming speed (*Polin et al., 2009*). Meanwhile PO swims forward using a rotary breaststroke with frequent episodes of ultrafast backward swimming (called 'shocks'), where all eight cilia undulate synchronously in front of the cell for up to $20\,\mathrm{ms}$ (*Wan and Goldstein, 2018*). During shocks cell speed can exceed $1\,\mathrm{mm\,s^{-1}}$.

Both species displayed stochastic transitions between different swimming modes (*Figure 2*). However, CR swims largely continuously albeit with noisy fluctuations in speed (fractal timeseries), PO exhibits excitable dynamics reminiscent of neuronal spiking, where bursts of high-speed movement are followed by periods of quiescence. This more episodic motility repertoire of PO is associated with

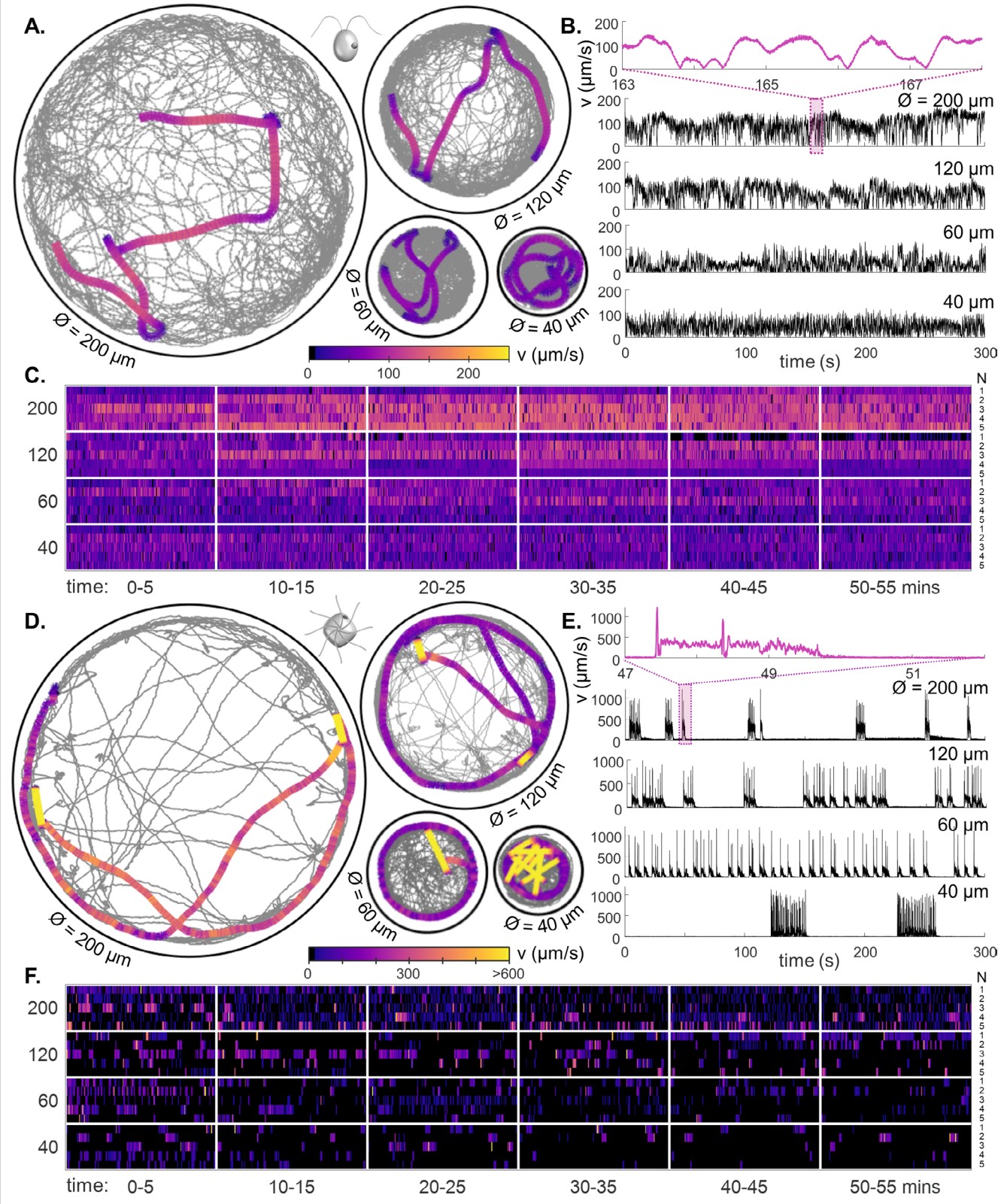

**Figure 2.** Behavioural ethograms of motile algae. (**A**) High-speed trajectories of single *C. reinhardtii* cells in circular traps of varying sizes (illustrating coverage over 5 min), overlaid with a 5-second representative trajectory (colour-coded by speed). (**B**) Instantaneous swimming speeds for the sample trajectories shown in (**A**). Inset: speed fluctuations over short timescales. (**C**) Heatmaps of cell swimming speed over time indexed by cell number N

*Figure 2 continued*

(rows), and ordered by trap size, using data from all experimental runs. (**D–F**) Similar, but for PO. All trajectories viewed from above the sample in the lab frame.

The online version of this article includes the following figure supplement(s) for figure 2:

**Figure supplement 1.** Summary violin plots of the speed data for different trap sizes, over the 6 timepoints (respectively, 0-5, 10-15, 20-25, 30-35, 40-45, and 50-55 min).

more sparse trap coverage over time (*Figure 2A and D*) (*Videos 1 and 2*). The distinction persists in the long-time single-cell heatmaps (*Figure 2C and F*). In the smallest traps, PO cells showed bursting dynamics - with clusters of shocks in quick succession (*Figure 2E*). Over 1 hr, CR motility remained largely unchanged, in PO a slight reduction in activity was observed in the smallest traps (*Figure 2— figure supplement 1*).

## Effect of physical confinement

We assayed four trap sizes (Ø = 40, 60, 120, 200 µm), varying from strong confinement where the cell is never more than 1.5 body lengths away from the boundary, to weak confinement where cells are free to roam along tight helices. Population-level trends and stereotyped behaviours were evident when we averaged trajectory statistics over all cells, across all timepoints.

Our findings are twofold. First, confinement affects cell swimming speed. As trap size is decreased, CR speeds are shifted towards lower values (*Figure 3A*, whereas in PO the characteristic speed and likelihood of runs are reduced (*Figure 3B*), compare location and width of the violin plot 'waist'). See also bivariate histograms of linear $v$ and angular speed $\Omega$ (*Figure 3—figure supplement 2*). Second, increasing confinement changes the nature of how cells explore the circular traps. To see this, we partition the arena (scaled by trap size) and compute the relative probability density of occupancy in each box. Averaging over all trajectories, we find that cells preferentially move within an annulus of the solid boundary (*Figure 3C, D* - shaded blue regions). To account for the 'arrow of time' in the movement history of a single individual (*Battle et al., 2016*; *Cammann et al., 2021*), we also compute for each trap size a time-averaged probability flux at each position (see Appendix 1). These steady-state fluxes have a heading and strength according to the most probable trajectory direction starting from the current position. Unexpectedly, single-cell trajectories display non-equilibrium flux loops resulting from persistent circling behaviour inside the traps (*Figure 3C, D*). Circulation is predominantly counter-clockwise (CCW), viewed from above in the lab frame.

To explain why the cell not only follows the trap boundary, but also how a preferred circling direction can emerge from a perfectly symmetrical trap, we turn to a minimal computational model (*Cammann et al., 2021*). For brevity, we consider here the case of a CR-like microswimmer only. Briefly, the swimmer is modelled as an active dumbbell with unequal-sized spheres, whose dynamics obey overdamped Langevin equations, with a forcing term to account for steric interactions with the solid boundary (see Appendix 1). We further assume cells swim with a constant speed of $v = 80$ µms⁻¹ in all traps. This results in boundary circulation behaviour without recourse to change the cell's internal motility state (*Ostapenko et al., 2018*). In domains with constant curvature (e.g. circular), however, macroscopic flux loops should not arise, due to equal probabilities of circling clockwise or counter-clockwise. A further source of asymmetry must be present to account for the experimental observations. Closer inspection of high-magnification videos of single CR cells revealed that the two flagella often lose synchrony, one flagellum consistently

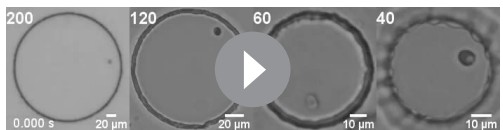

**Video 1.** Example videos showing single *Chlamydomonas reinhardtii* cells trapped by droplet microfluidics inside circular arenas of different diameters (Ø = 200, 120, 60, 40 µm), under red light illumination.

https://elifesciences.org/articles/76519/figures#video1

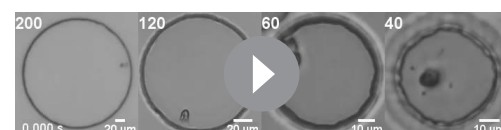

**Video 2.** Example videos showing single *Pyramimonas octopus* cells trapped by droplet microfluidics inside circular arenas of different diameters (Ø = 200, 120, 60, 40 µm), under red light illumination.

https://elifesciences.org/articles/76519/figures#video2

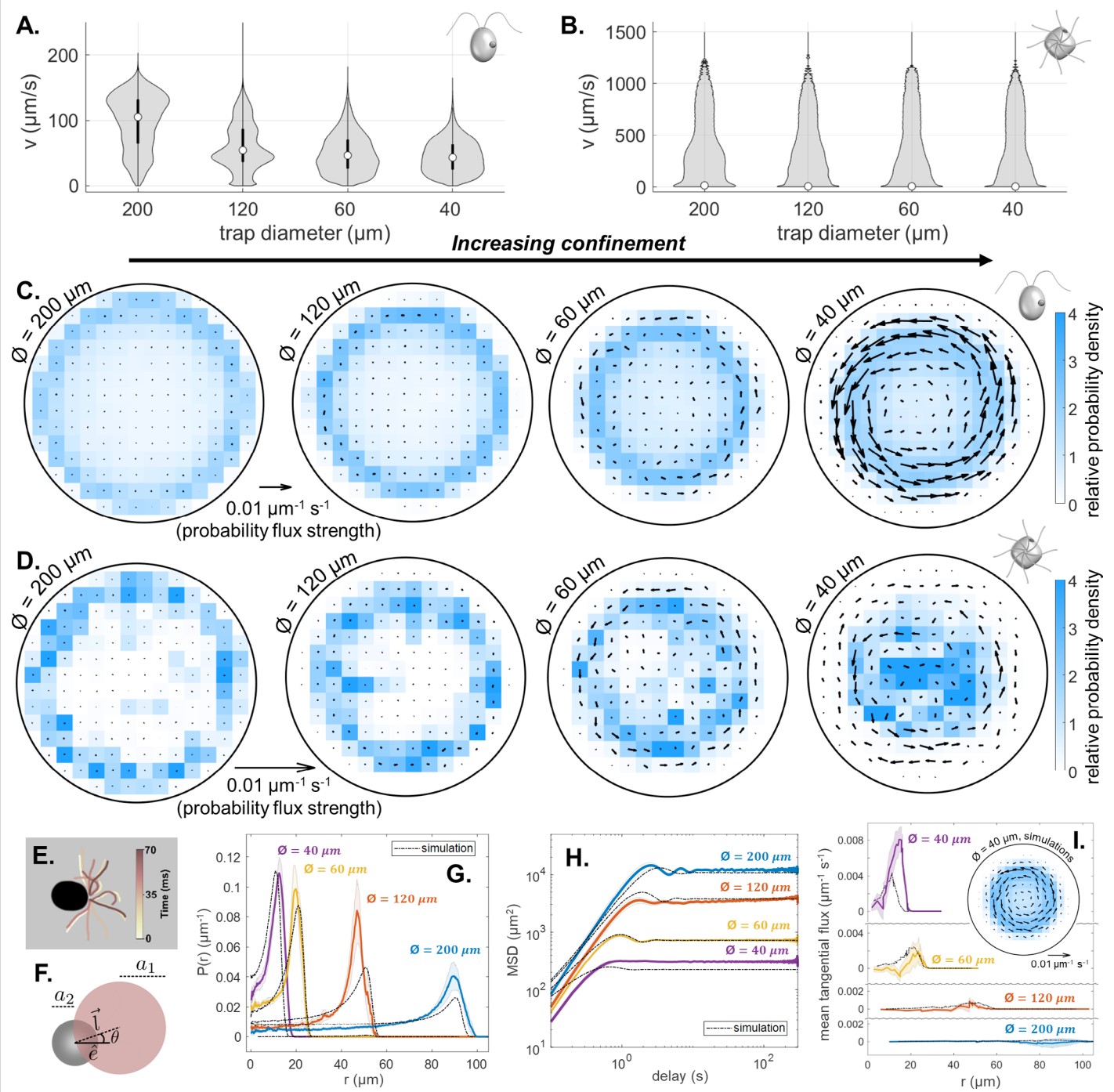

**Figure 3.** Effect of trap size on single cell motility. (**A,B**) Violin plots of pdfs of speed ($v$) across all cells, and all time points, for CR and PO. (Note the scaling of the pdfs is linear for CR, but logarithmic for PO.) (**C,D**) Time-averaged heatmaps of the cells' centroid positions, for each of the four different trap sizes, overlaid with arrows showing the direction and strength of steady-state non-equilibrium fluxes computed from trajectory statistics, for CR and PO. (**E**) Traced cilia waveforms for CR showing the asymmetric beat pattern of the two cilia under red light. (**F**) The asymmetric dumbbell model used for CR with a smaller circle (radius $a_2$) for the cell body and a larger circle (radius $a_1$) representing the area covered by the cilia beat, which is offset from the swimming direction $\hat{e}$ by angle $\theta$. Comparison of the experimental mean and standard error ($N = 5$) of the radial probability densities $P(r)$ (**G**), mean-squared displacement curves (**H**) and mean tangential flux (**I**) with the simulation results for each trap size. The inset in (**I**) shows the time-averaged heatmap of centroid position overlaid with arrows showing the probability flux results for the simulation of the 40 μm diameter trap.

The online version of this article includes the following figure supplement(s) for figure 3:

*Figure 3 continued on next page*

*Figure 3 continued*

**Figure supplement 1.** The experimental mean and standard error (N = 5) of the radial probability densities $P(r)$ (**A**) and mean-squared displacement curves (**B**) for each of the trap sizes, for PO.

**Figure supplement 2.** Bivariate histograms of linear and angular speed ($v$, $\Omega$) across all cells, and all time points for CR (**A**), and PO (**B**).

beating faster than the other leading to bilateral asymmetry (*Figure 3E*). Motivated by this, we modify the dumbbell model so that the two spheres are misaligned with an offset angle $\theta$ (*Figure 3F*). For a small offset $\theta = 1.5\,^\circ$ (where $\theta > 0$ is the CCW direction), we simulated single-swimmer trajectories for the four trap sizes and evaluated the long-time behaviour. Radial probability densities, mean-squared displacement (MSD) profiles, and mean tangential fluxes of the simulated tracks agree rather well with measured data (*Figure 3G-I*). In the simulations, chiral fluxes became more prominent with increasing confinement, consistent with the data (*Figure 3I*).

## Effect of white light stimulation

Next we explore the effect of an orthogonal parameter on motility - illumination wavelength. We imaged the samples with broad-spectrum white light (WL) instead of red light (RL), at an intensity expected to influence phototaxis in CR, but not sufficient for photoshock. For two trap sizes, $\varnothing = 120$ μm and 60 μm (again $N = 5$ per trap size), we again recorded motility over 1 hour (*Videos 3 and 4*).

Some single-cell heterogeneity is evident (*Figure 4*). At early times CR cells swim faster in WL compared to RL in the same trap sizes at equivalent times, suggesting a photokinetic response (*Figure 4A*, *Figure 4—figure supplement 1A*). A decline in motility is observed across all sampled cells after $\sim 45$ min, especially in the smallest 60 μm traps (note wide 'waists' at zero speed). All cells in the 60 μm traps were completely immotile after $\sim 50$ min. Similarly, PO became increasingly quiescent and immotile over time. By 1 hr, all cells stopped moving (*Figure 4B*), and in some cases deformation of cell shape and even deciliation was noted at the final time point. This was not seen in RL (*Figure 2*, *Figure 2—figure supplement 1*). Comparing PO speeds in WL and RL at early times, maximum speeds during ciliary reversals or shocks are unchanged, but their distributions are skewed toward higher speeds due to increased likelihood of forward runs ($300 - 400$ μms$^{-1}$) (*Figure 4—figure supplement 1B*).

As before, we compute the spatial occupancy of single-cell tracks and their associated steady-state (time-averaged) fluxes (*Figure 4C*). Again, directed flux loops emerged, due to circling behaviour, most prominently for CR. For both species, the flux strengths are larger and the circulation patterns more ordered in WL than in RL. The predominant sense of circulation also switched from CCW to clockwise (CW).

To explain this, we return to our minimal model, and again consider a CR-like swimmer for simplicity. We first increase the swimming speed from $v_0 = 80$ μms$^{-1}$ to $v_0 = 100$ μms$^{-1}$, to reflect the increase in speed in WL at early times compared to RL. As before, we hypothesize that biflagellar synchrony may be key. High-magnification imaging of CR swimming in WL revealed a dramatic improvement in flagellar synchrony, consistent with previous observations (*Wan et al., 2014*). Accordingly, we update the offset angle from $\theta = +1.5\,^\circ$ in RL to $\theta = -1\,^\circ$ in WL. These values are illustrative, as there is no clear mapping between flagellar synchrony and shape offset. With these two simple modifications, the model reproduces both the increase in flux strength and reversal in flux direction in WL (*Figure 4F*

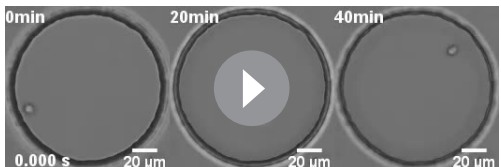

**Video 3.** A single *Chlamydomonas reinhardtii* cell is trapped inside a circular arena of diameter (∅ = 120 μm), under white light illumination. Videos show different timepoints for the same individual.
https://elifesciences.org/articles/76519/figures#video3

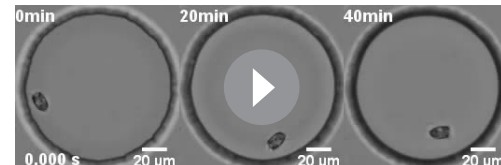

**Video 4.** A single *Pyramimonas octopus* cell is trapped inside a circular arena of diameter (∅ =120 μm), under white light illumination. Videos show different timepoints for the same individual.
https://elifesciences.org/articles/76519/figures#video4

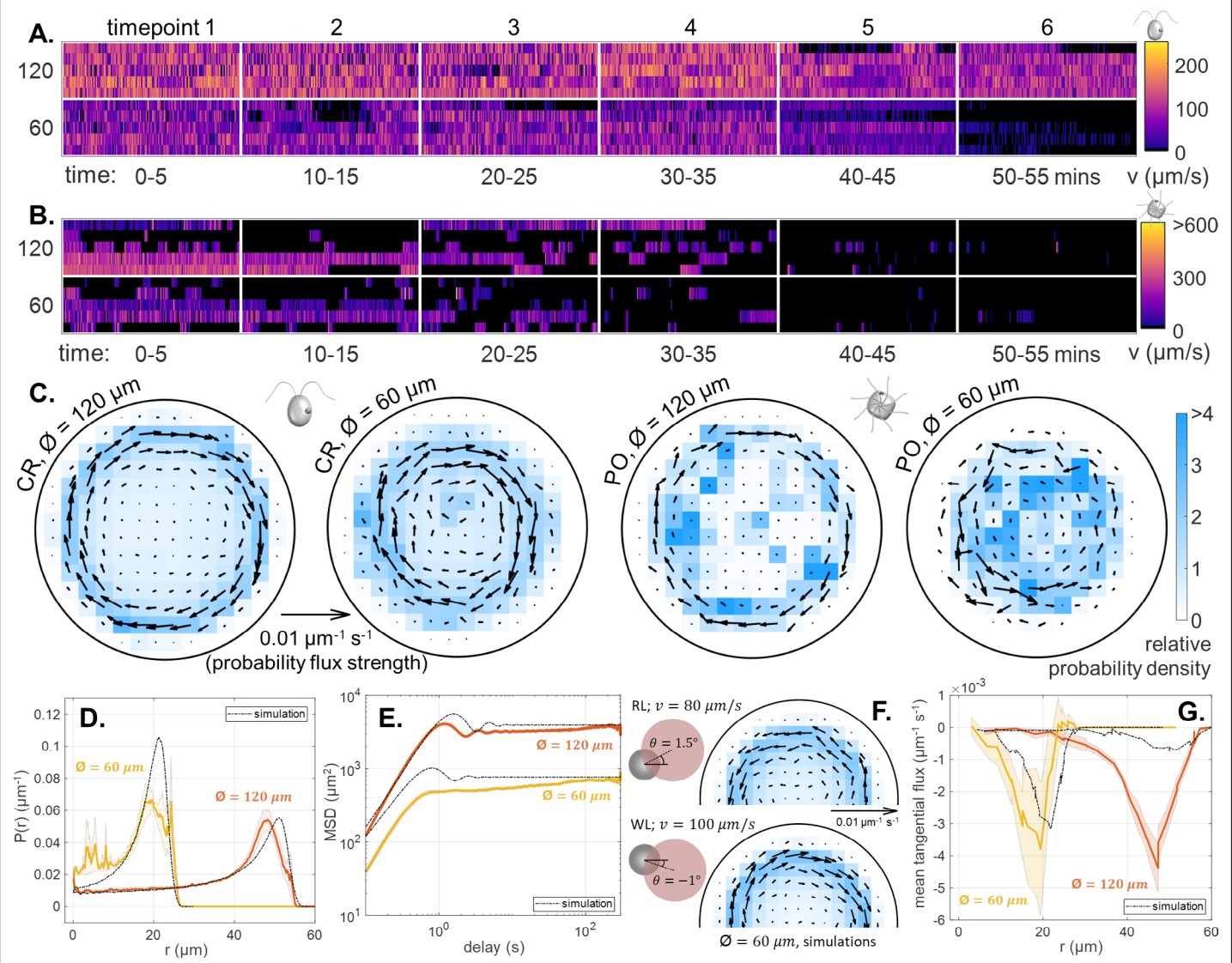

**Figure 4.** Effect of white light on single cell motility. (**A,B**) Heatmaps of cell swimming speed over time, for CR and PO. (**C**) Time-averaged heatmaps of the cells' centroid position overlaid with arrows showing the direction and strength of the probability flux computed from trajectory statistics for both trap sizes under white light conditions. Comparison of the experimental mean and standard error ($N = 5$) of the radial probability densities $P(r)$ (**D**) and mean-squared displacement curves (**E**) with the simulation results for each trap size. (**F**) An illustration of the different dumbbell model parameters used for RL and WL conditions together with semi-circles showing the time-averaged heatmaps of the centroid position overlaid with probability flux arrows for the 60 μm diameter trap size. A change in the direction of the offset angle $\theta$ changes the direction of the non-equilibrium flux. (**G**) Comparison of the simulations results with the experimental mean and standard error ($N = 5$) of the mean tangential flux as a function of radial displacement for each trap size.

The online version of this article includes the following figure supplement(s) for figure 4:

**Figure supplement 1.** Violin plots of the pdfs of speed over time for all cells, for CR (**A**) and PO (**B**).

*and G*). The simulations can be extended to account for more complex motility patterns (e.g. episodic swimming in PO), or a variable swimming speed.

## Behaviour is compressed into three motility macrostates

How must the dynamics of individual locomotor appendages change, to explain the above whole-cell behaviours? How does this depend on the species? To answer these questions, we use long-time statistics to provide a robust measure of behaviour that is independent of a cell's physical environment. We first observe that the motility space is low-dimensional, comprising only a trio of movement

states (*Figure 5* and Appendix 1). Both species exhibit a quiescent or 'stop' state in which the cell body and cilia exhibits minimal movement, a 'run' state associated with constant speed and smooth forward swimming, and a 'transitional' state for re-orientations. This is analogous to prokaryotic strategies such as run-and-tumble in *E. coli* (*Berg, 2003*; *Perez Ipiña et al., 2019*), or the run-reverse-flick in *V. alginolyticus* (*Son et al., 2015*).

We can decompose long-time tracks into discrete timeseries of these states (labelled $i = 0, 1, 2$) according to a heuristic approach based on direct observations of ciliary beat patterns and associated trajectory parameters such as linear $v$ or angular velocity $\Omega$. At time $t$, the cell is in state $S(t)$. For CR, we define 'runs' ($S = 1$) to be when the two cilia engage in synchronous breaststrokes (high $v$, low $\Omega$), 'tumbles' ($S = 2$) when the cilia lose synchrony (low v, high $\Omega$), and 'stops' ($S = 0$) where $v < v_c$ for a threshold speed $v_c$ (*Figure 5A–D*). Similarly, the behaviour of PO can also be mapped to a tripartite run-stop-shock repertoire (*Wan and Goldstein, 2018*), where PO 'shocks' are analogous with CR 'tumbles'. 'Runs' ($S = 1$) result from coordinated breaststrokes involving 8-cilia (*Wan, 2020*), corresponding to moderate $v$ and low $\Omega$. 'Shocks' ($S = 2$) occur when all eight cilia switch to a symmetric beat and undulate in front of cell, producing rapid backward swimming followed by reorientation (high $v$ and high $d\Omega/dt$). Finally, 'stops' ($S = 0$) are where $v < v_c$ for a threshold speed $v_c$ (*Figure 5H–K*). We note that for both species while coordinated strokes produce straight swimming across the trap interior ('runs'), curvature-guided interactions with the wall lead to circling behaviour ('runs' along the boundary); both are considered 'runs' here (*Figure 5B and I*).

This discrete state representation allows us to quantify how *sub-cellular* dynamics change over time or in response to environmental cues. We can estimate for each cell and for each assayed condition (see Appendix 1) the expected probability of being in state $i$ ($p_i$), the list of sojourn times in state $i$ ($T_i$), survival probabilities $P(T_i > \tau)$ and expected residence times $\langle T_i \rangle$ in state $i$, as well as the pairwise state transition probability ($p_{ij}$) and the transition rate (number of transitions per unit time) $q_{ij}$ from state $i$ to $j$. Together, these rate constants specify a unique network akin to a chemical reaction network. An example from the 120 μm traps is shown (*Figure 5E and L*). See *Figure 5—figure supplement 1* for results in the 60 μm traps.

In terms of state probabilities for CR in the 120 μm traps (*Figure 5G*): $p_1^{WL} > p_1^{RL}$ (runs more likely in WL), $p_2^{WL} < p_2^{RL}$ (tumbles less likely in WL), but $p_0^{RL} \approx p_0^{WL}$ (both equally likely to stop swimming by 1 hr). WL also increased the tumble→run transition rate ($q_{21}$) and decreased the run→tumble transition rate ($q_{12}$) (*Figure 5G*), Empirically, this means that CR performs more frequent and longer runs in WL compared to RL. For PO, the difference between RL and WL is more obvious, see for example the run-state survival probability $P(T_1 > \tau)$ (*Figure 5M*). State probabilities $p_{0,1,2}^{RL}$ remained largely constant over 1 hr, but $p_{1,2}^{WL}$ decreased over time (runs and shocks became less likely) while $p_0^{WL}$ increased (stop more likely) (*Figure 5N*). Initially $p_1^{WL} > p_1^{RL}$, but after the 4th timepoint (about 40 min) this drops to below the corresponding RL value. We see that $q_{12}^{WL} < q_{12}^{RL}$ (run to shock transitions are much less likely in WL), and $q_{10}^{WL}$ increased over time so that run→stop transitions became more likely. This means that in WL, PO runs are longer and more likely than in RL, but cells gradually lost motility over time.

## Algal cell motility is light-switchable

Next we perform a dynamic assay to see how the *same* individual responds to photostimulation, and if such responses are reversible. Single CR or PO cells were trapped in 120 μm wells (again $N = 5$ per trap size) and imaged continuously for 20 minutes in the following sequence: (1) 5 minutes in RL (RL1), (2) 5 min photostimulation in WL (WL1), (3) 10 min back in RL (RL2). For both species, motility patterns in RL1 and WL1 are similar to previous results under constant RL or WL illumination. As before, single-cell heterogeneity is observed (*Figure 6A and B*).

Both species display a reversible but asymmetric photokinetic response to WL1. Response to RL → WL (step-up) is fast and occurs within seconds, but recovery after WL → RL (step-down) is slow and persists for several minutes into the RL2 phase (*Figure 6C and D*). Most cells swim faster with longer runs in WL1 period than in the preceding RL1 period. In CR this is primarily by increasing the speed of prospective runs (*Figure 6E*), but in PO this is by reducing the likelihood of state transitions, particularly of shocks immediately in response to step-up (*Figure 6D and F*). The suppression of shocks in PO is consistent with previous results in constant WL. Cells gradually *adapt* by increasing the transition frequency over the remainder of WL1, and well into RL2.

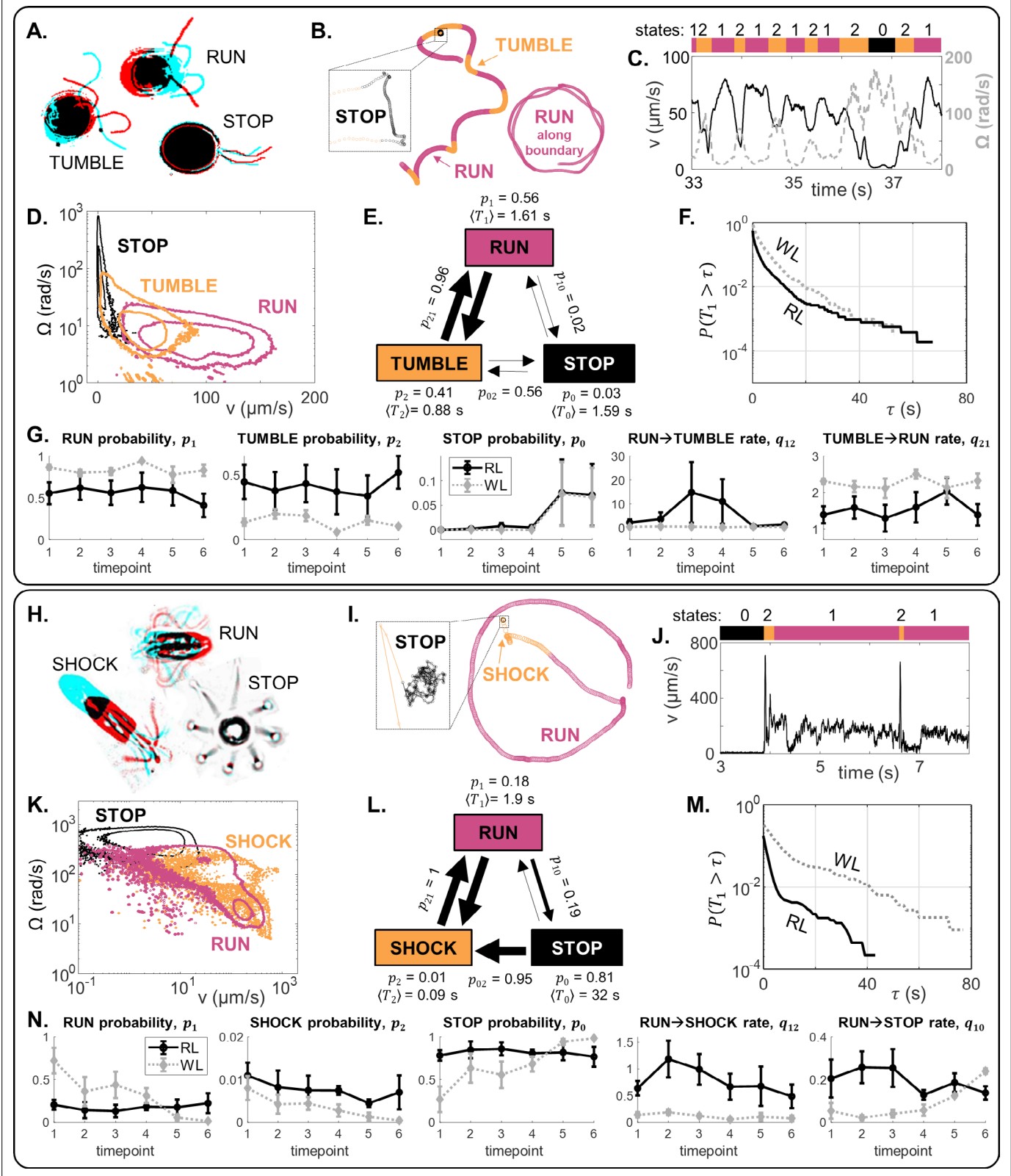

**Figure 5.** The locomotor behaviour of single-cells is controlled by a trio of motility macrostates and their transition probabilities. (**A–G**) For CR, distinct ciliary beat patterns and coordination states (**A**) produce different swimming trajectories (**B**), and a discrete sequence (**C**) of states (run, stop, tumble). For cells in 120 μm traps, RL, the states occupy distinct regions in $(v, \Omega)$-space (**D**), and a time-averaged reaction network with characteristic transition

*Figure 5 continued on next page*

*Figure 5 continued*

parameters (**E**). Network parameters are different in WL for the same trap size, as evidenced in (**F**): survival probability of run states, and (**G**): mean and standard error (N = 5) for selected state probabilities and transition rates. (**H–N**) Similarly for PO.

The online version of this article includes the following figure supplement(s) for figure 5:

**Figure supplement 1.** Survival probabilities and selected reaction network parameters comparing the 120 μm and 60 μm traps in RL and WL for CR (**A–B**) and PO (**C–D**).

Thus, algal swimming motility is light-switchable in the presence of short-lived stimuli, which induce perturbations to the cell's internal motility state network. Cells retained a *memory* of the stimulation (WL1) for longer than the stimulus duration itself. Analysing 'run' phases exclusively to extract the most prominent frequencies from the helical trajectories (see Appendix 1), we further deduce that CR swims faster by increasing the frequency *and* synchronicity of ciliary beating (from $52 \pm 4$ to $62 \pm 3$ Hz), in response to step-up from RL1 to WL1 (*Figure 6G*). A similar analysis of PO trajectories yielded significant cell-to-cell variability but no clear frequency signatures that could be correlated with ciliary beating (*Figure 6H*). This is likely due to the unique coordination of the 8 cilia that suppresses the helicity of trajectories (*Wan, 2020*).

## Effect of chemical stimulation

In many protists, ciliary motility is modulated by chemical sigalling, including $Ca^{2+}$-influx through ion channels, suggesting a conserved Ca-dependent mechanism (*Inaba, 2015*; *Wan and Jékely, 2021*). Ionic fluxes elicit transitions in cilia beating, waveform and swimming behaviour (*Beck and Uhl, 1994*; *Kung and Naito, 1973*; *Brette, 2021*; *Geyer et al., 2022*). In *Paramecium*, $Ca^{2+}$, and $K^+$ ions are major regulators of motility, including transitions between forward and backward swimming (*Kunita et al., 2014*). We hypothesize that motile algae also possess a behavioural response to changes in external ion concentrations, and confirmed this in PO with a bulk motility assay (see Materials and methods). The introduction of a droplet pre-loaded with 50 mM KCl to a suspension of PO cells produces perturbed swimming (*Figure 7A*).

Traditional, bulk assays do not allow path-sampling of single-cell trajectories, nor do they have the ability to track in real-time how a single cell responds to chemical perturbations. We solve this by designing a microfluidic device to deliver controlled fusion of two droplets (*Figure 7B*). Using a cross-junction microfluidic chip, we generated droplets in alternation that contain either a 10 mM KCl solution or cell suspension. Pairs of droplets were then trapped in downstream doublet wells. We then identified droplet pairs comprising one with a trapped PO cell, and another containing the 10 mM KCl solution spiked with 1 μM of fluorescein which acted as a reporter. Fusion is triggered by starting the flow of 40% perfluoro-1-octanol (PFO) in HFE (a competing surfactant); see Materials and methods. Two such droplets fuse within 1 or 2 ms. Note that trap-occupancy is stochastic: not all droplet pairs have the target composition, since droplet generation does not alternate perfectly between the two channels. Doublet wells with both droplets containing KCl, or both droplets containing cell suspension, were not imaged (*Figure 7B*, inset).

We compare the motility of cells in these doublet traps in the presence or absence of the chemical perturbation. Control experiments were performed by substituting the KCl medium with normal culture medium (without cells). Prior to fusion, cell motility was similar in both cases (*Figure 7C*). After fusion, cells paired with a KCl droplet immediately showed erratic swimming, followed by gradual decay in motility over time (see *Video 5*), which is likely a combined effect of an increase in $K^+$ and decrease in $Ca^{2+}$ concentration (*Table 1*). Significant changes in the morphology and tortuosity of swimming trajectories are also observed after fusion (*Figure 7D and E*). Further examples of this response are presented in *Figure 7—figure supplement 1*.

## Discussion
### Stereotypy and the 'arrow of time' in long-term single cell behaviour
The emerging field of computational ethology seeks to track and annotate the complex behaviours of organisms (*Han et al., 2018*; *Yaski et al., 2011*; *Yemini et al., 2013*; *Stephens et al., 2011*). Due to their small size, variable morphology, and high-speed movement, detailed analysis of the

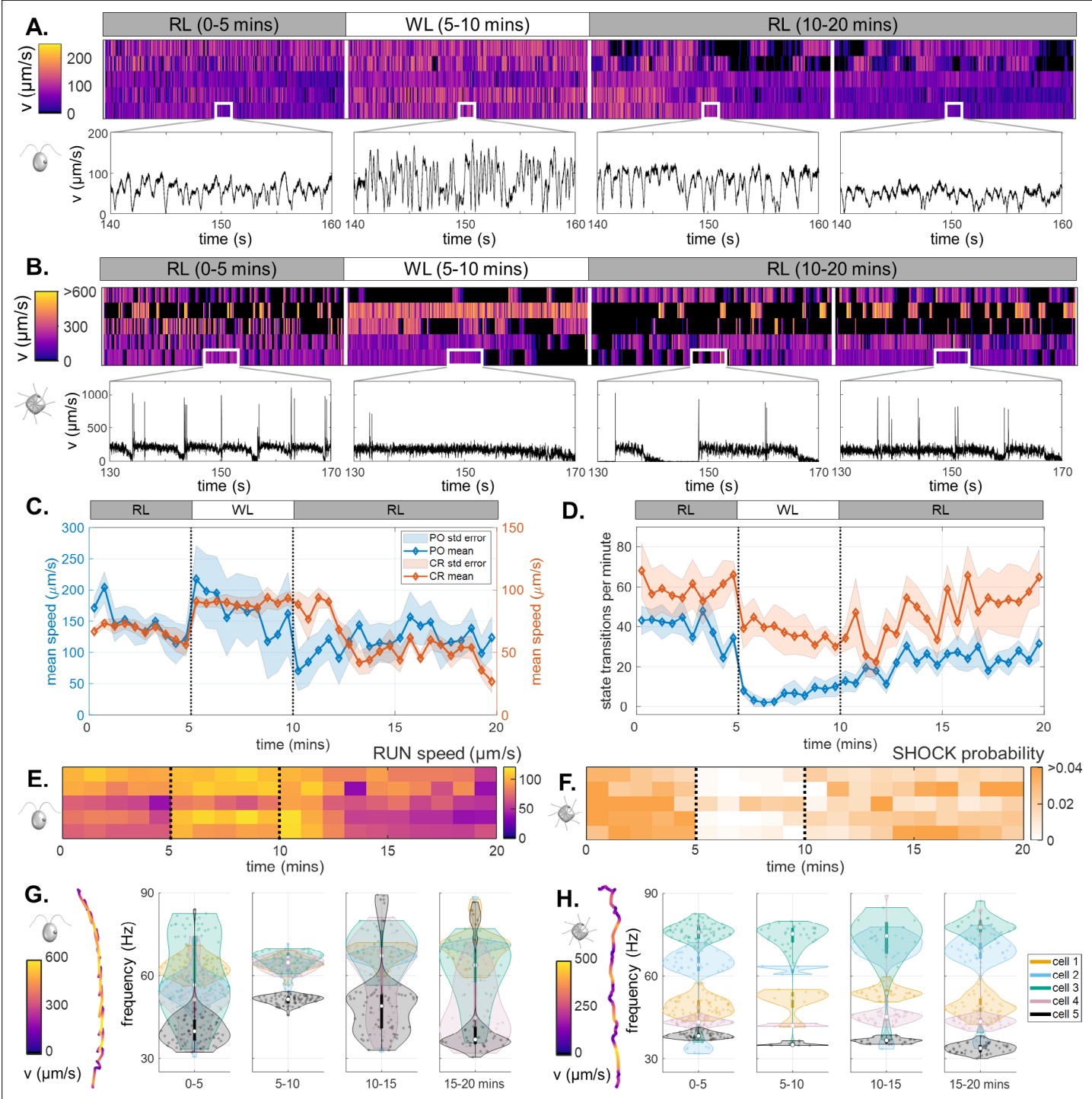

**Figure 6.** Light-switchable motility of green algae. (**A**) Heatmaps of CR swimming speed over the course of a 20-min photoresponse assay, insets show short, representative timeseries in each phase. (**B**) The same for PO. (**C**) Population-averaged swimming speed, over time. (**D**) Population-averaged rates of transitions between motility states, over time. (Shaded regions are standard errors.) (**E**) Mean speed of 'runs' in CR, over time. (**F**) Probability of 'shocks' in PO, over time. Violin plots of the most prominent frequencies extracted from helical trajectories (inset), for CR (**G**) and for PO (**H**). For CR, this corresponds to the ciliary beat frequency.

The online version of this article includes the following figure supplement(s) for figure 6:

**Figure supplement 1.** Light switchable states data over time for individual cells of CR (**A**) and PO (**B**).

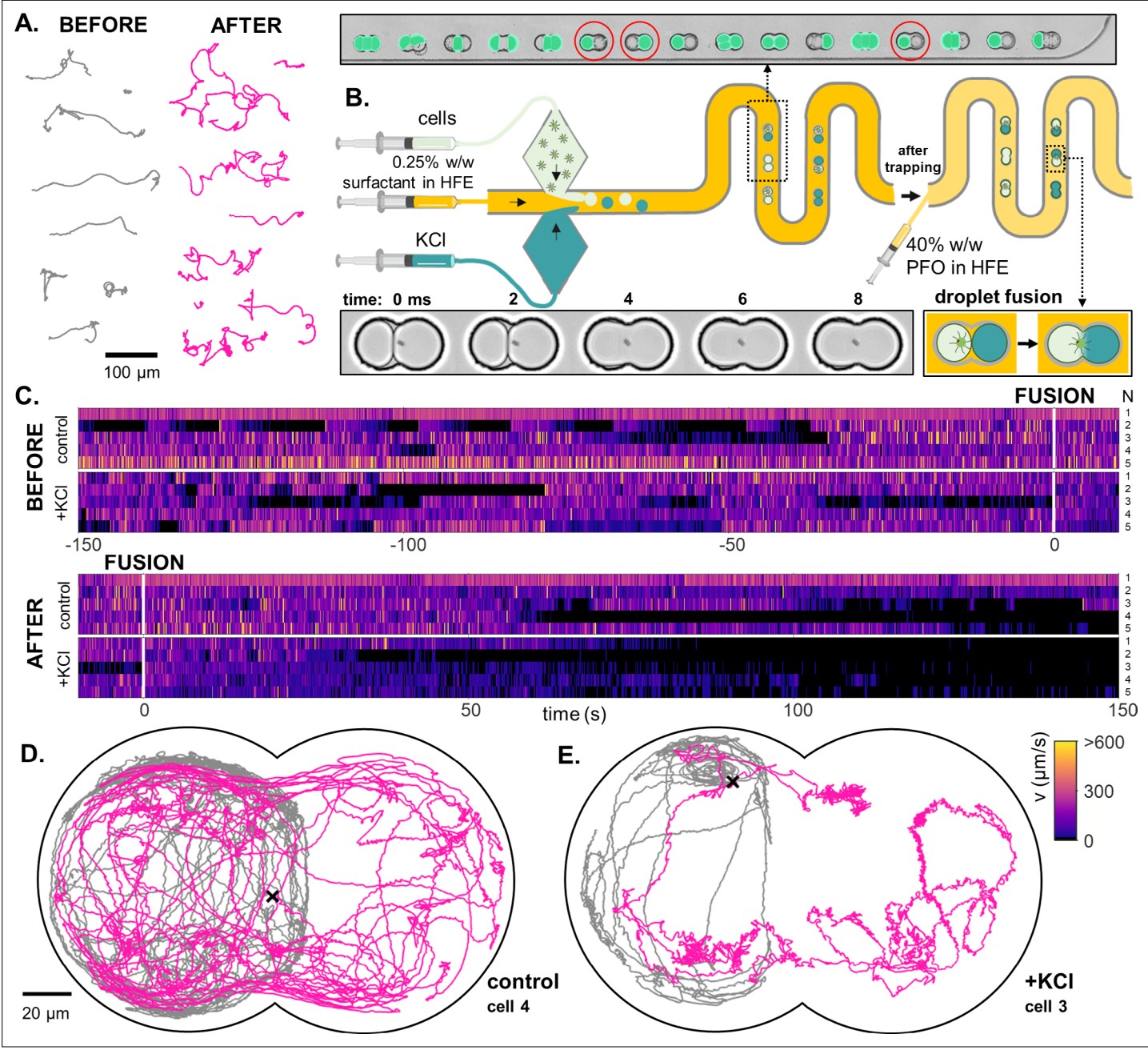

**Figure 7.** A novel droplet-fusion assay for querying single-cell motility responses to rapid chemical perturbation. (**A**) Example tracks from a bulk motility assay (see Materials and methods), immediately *before* ('normal motility phenotype') and *after* addition of 50 mM KCl to the medium ('chemically perturbed phenotype'). (**B**) Schematic of the serpentine microfluidic device for generation and pairing of droplets - one containing the trapped cell, and the other loaded with 10 mM KCl (inset, red circles). Fluorescein was added to the KCl phase to identify paired droplets. Fusion was then induced by replacing the 0.5% surfactant in HFE with 40% 1 H,1H,2H,2H-Perfluoro-1-octanol (PFO) in HFE. Two such droplets merge within 1 or 2 ms (inset). (**C**) Heatmaps of single-cell swimming speed, before and after the moment of fusion to a droplet containing KCl, compared to a control pair where KCl was replaced with inert culture medium. An example single-cell trajectory 50 s before (grey) and 50 s after (pink) droplet fusion, for a control (**D**) and a KCl pair (**E**).

The online version of this article includes the following figure supplement(s) for figure 7:

**Figure supplement 1.** More sample trajectories for control (**A**) and KCl (**B**) droplet pairs.

**Video 5.** Example video showing a single *Pyramimonas octopus* cell before and after fusion with KCl solution, under red light illumination.
https://elifesciences.org/articles/76519/figures#video5

behavioural patterns of microorganisms is technically challenging. Here, we leveraged droplet microfluidics to track the swimming trajectories of microbes for long periods of time. Comparing a freshwater biflagellate with a marine octoflagellate, we quantified how single-cells respond to a number of controlled environmental stimuli (including mechanical, light and chemical).

We showed that the behavioural space of roaming microbes comprise highly stereotyped movements, and proposed a new paradigm for phenotyping microbial motility in terms of a minimal set of three motility macrostates (runs, stops, tumbles/shocks). This inherent low-dimensionality has been observed in other organisms (*Larson et al., 2022*; *Jordan et al., 2013*; *Tsang et al., 2018*) and artificial microswimmers (*Leoni et al., 2020*; *Hokmabad et al., 2021*). Over long times, CR swimming is fairly continuous, yet PO transitions between bursts of fast swimming and lengthy periods of quiescence, reminiscent of 'wakeful' versus 'sleep states' observed in animals (*Lawler et al., 2021*).

Importantly, our long-term behavioural recordings retained the 'arrow of time' (*Gnesotto et al., 2018*). This gives us access to key information that may be hidden by traditional, bulk-averaged measurements, such as chiral movements in confined spaces (see next section). From these ultralong timeseries, we quantified how behaviour changes at the sub-cellular level. Transition rates and probabilities between states were shown to not only change in time but also in response to environmental stimuli. Such timeseries can also be mined to infer non-equilibrium entropy production (*Skinner and Dunkel, 2021*; *Wan and Goldstein, 2018*). These species-specific movement signatures reflect different intracellular processes for controlling the algal motility apparatus (*Guo et al., 2021*), and could be key to the emergence of cognition and environmental responsiveness (*Kunita et al., 2016*; *Wan and Jékely, 2021*). Microbes inhabiting different ecological niches (e.g. freshwater or marine) likely evolved divergent strategies for sensorimotor control, despite all relying on the same organelle for motility. Future work should seek to map the space of ciliary waveforms, itself low-dimensional (*Geyer et al., 2022*), to whole-cell swimming behaviours.

## Flux loops without curvature gradients

Microbes often interact with their physical environment, particularly with interfaces and boundaries. Interactions are commonplace in natural settings with exposed surface features or other heterogeneous structures, for example soil, foam, or particulate matter (*Théry et al., 2021*; *Kantsler et al., 2013*; *Souzy et al., 2022*). Here, we engineered PDMS chambers with precise shapes to explore how confinement affects cell motility. In red light, cells tolerated long-periods of confinement with little change to their overall motility. We also discovered a chiral circling behaviour, confirmed by flux representations of the time-averaged trajectories. While confinement has been shown previously to stabilise macroscopic chiral movement in swimmer suspensions (*Wioland et al., 2013*; *Beppu et al., 2021*), here chirality has emerged in the travel history of a *single cell*. This implicates a distinct route towards a chiral non-equilibrium steady-state that has not been reported previously.

That the sense of chirality also depends on light is strong indication that it arises from a mechanism under cellular control. We outlined a model for how this can occur, based on a minimal representation of a CR-like cell as a dumbbell with fore-aft asymmetry (*Cammann et al., 2021*). Such a swimmer will align with solid boundaries in a curvature-dependent manner, as shown previously (*Ostapenko et al., 2018*). In a circular trap

**Table 1.** A comparison of the concentrations of key ions (in mM), before and after paired droplet fusion.

| Ion | Pre-fusion | Post-fusion (with 10 mM KCl solution) |
|---|---|---|
| $Na^+$ | 395 | 197 |
| $K^+$ | 8.33 | 9.17 |
| $Ca^{2+}$ | 8.75 | 4.38 |
| $Mg^{2+}$ | 44.5 | 22.3 |
| $Cl^-$ | 459 | 230 |
| $SO_4^{2-}$ | 23.6 | 11.8 |
| $NO_3^-$ | 1.1 | 0.6 |
| $HPO_4^{2-}$ | 0.1 | negligible |

(constant curvature), it should not produce flux loops on average (*Cammann et al., 2021*). We find that breaking of the dumbbell swimmer's left-right symmetry was sufficient to produce trajectories with CW or CCW bias, depending on the sign of the offset $\theta$. This is supported by data showing the bilateral *flagellar dominance* (asymmetry between *cis* and *trans* flagella) to be tunable by light. This is also important for helical swimming and klinotaxis (*Cortese and Wan, 2021*; *Leptos et al., 2022*). Thus, even minor cell-internal asymmetries (here represented by a very small $|\theta|$) are sufficient to break macroscopic symmetry. Further particulars of cell shape (*Lushi et al., 2017*; *Zaferani et al., 2021*), wall interactions (*Lauga et al., 2006*; *Spagnolie et al., 2017*), or cilia mechanosensitivity (*Wan and Goldstein, 2018*) may be needed to account for the more complex swimming repertoire of PO.

## Light-dependent algal motility and phototoxicity

We also found that motility parameters in RL are more stable than in WL, for both species. Short periods of WL exposure increases the average run-speed of CR by increasing ciliary beat frequency, as found previously in micropipette-immobilised cells (*Rüffer and Nultsch, 1990*). In contrast, WL decreases the frequency of gait transitions in PO, producing longer stops, longer runs, and fewer shocks. Prolonged or excessive exposure (> 40 minutes) to WL eventually led to reduced or cessation of motility in both species. In CR this is likely due to WL-induced adhesion (*Kreis et al., 2018*). In PO, there is more evidence of irreversible photodamage. Increasing confinement (smaller traps) decreased the time taken to reach this state, suggesting that motility deterioration may be due to buildup of some metabolite or excretion from the trapped cell over time (*Boedicker et al., 2009*).

Shorter periods ($\sim$ 5 min) of WL-stimulation produced reversible changes in motility in the algae (*Figure 6*), again with species-dependent photokinetic responses occurring at the single cell level. While CR responded largely by modulating run speed, PO modulated the balance of motility macro-states (e.g. suppressed shocks). These distinct behaviours offer intriguing prospects for synthetic biology and bioengineering, such as light-guided patterning of microbes (*Frangipane et al., 2018*; *Bittermann et al., 2021*). It will be interesting to combine our assay with genetic perturbations (particularly in CR) to reveal the regulatory pathways responsible for photosynthesis, phototransduction, and phototaxis.

## Future prospects of droplet microfluidics for assaying cell motility

Micro-encapsulation provides unique functionalities that enable controlled, on-demand creation and manipulation of a cell's local environment (*Son et al., 2015*). A microfluidics-based pipeline allows us to stably trap motile cells, to establish their unique behavioural signatures. We use multilayer devices and static droplet trapping wells (*Fradet et al., 2011*) to keep droplets stable for hours, and isolated from possible fluctuations in the continuous phase (*Han et al., 2020*). This allows us to achieve long-time imaging, and reliable control over trap size, shape, and geometry. The trapping arrays we used maximise the probability of obtaining multiple singly-trapped cells in a single experiment (only limited by Poisson statistics), thus increasing analytical throughput. Our current design focuses on imaging a small number of traps at a time. Simultaneous, high-throughput imaging of multiple cells in multiple wells is also possible, but at the detriment of spatial resolution and thus may not be sufficient for identifying motility state or behavioural transitions.

Our work paves the way for new applications and opportunities for designing diagnostic tools in the absence of molecular tests. Microfluidics-based assays enable the detection of single-cell motility signatures and any heterogeneity in microbial behaviour. Phenotypic diversity is critical in microbial ecosystems, where they may give rise to distinct selection pressures and antimicrobial resistance (*Dhar and McKinney, 2007*). Our droplet fusion device represents a highly novel solution for assaying fast cell-environment interactions and responses to chemical perturbations (*Figure 7*). Contrary to traditional assays that use flow channels to deliver concentration gradients, our design applies real-time control and localised perturbations to single microbes. Further developments and extensions are on the way, including augmentation of on-demand single cell encapsulation with active cell sorting (*Anagnostidis et al., 2020*; *Howell et al., 2022*). The integration of lab-on-chip technologies, high-speed microscopy and computer vision has significant potential for reconstructing the species-specific sensorimotor pathways of microorganisms, and revealing their response thresholds to dynamic environmental perturbations.

## Materials and methods
### Cell culturing and maintenance

*C. reinhardtii* cultures were prepared from axenic plates of the CC125 WT strain (Chlamydomonas Center). Individual colonies were transferred to $25\,cm^3$ volumes of Tris-minimal media. Liquid cultures were grown under a 14/10 light-dark cycle at 21°C and 40% humidity, with constant shaking at $110\,rpm$. Liquid cultures were sub-cultured when they approached the end of the exponential phase. For motility experiments, second- or third- generation cultures were harvested in late-exponential phase (6-9 days after inoculation). Cell density was measured as ~1 × $10^6$ cells per mL. Cells were centrifuged at 100 g for 10 min, and then concentrated 10-fold. The cells were then left in darkness for a minimum of 30 min to dark-adapt them before each experimental run.

We prepared our PO cultures from axenic liquid cultures of the WT of the species *P. octopus* (NIVA/NORCCA). 200 µL of axenic culture was transferred to $25\,cm^3$ of TL30 media. Cells were grown under continuous illumination at 21°C and 40% humidity, without shaking. For motility experiments, cultures were harvested during the latter half of the exponential phase (20-30 days after inoculation). Cell density was measured at ~3 × $10^3$ cells per mL. Cells were centrifuged at 100 g for 10 min, and then concentrated 10-fold. The cells were then left in darkness for a minimum of 30 min to dark-adapt them before each experimental run.

### Microfluidic chip fabrication

The devices were designed with CAD software (DraftSight, Dassault Systems) and fabricated following classical soft-lithography procedures by using a high-resolution acetate mask (Microlithography Services Ltd.). Negative photoresist SU-8 3025 (MicroChem, Newton, MA) was spin-coated onto clean silicon wafers to a thickness of 10 µm, patterned by exposure to UV light through the photomask and hard bake at 95°C for ~7 minutes (*Xia and Whitesides, 1998*). Prior to development through immersion in propylene glycol monomethyl ether acetate (PGMEA, Sigma-Aldrich), a second layer of SU-8 3025 at 20 µm in height was spin-coated, UV exposed and hard baked (95°C, ~7 minutes) for the development of the trapping arrays. Uncured polydimethylsiloxane (PDMS) consisting of a 10:1 polymer to cross-linker mixture (Sylgard 184) was poured onto the master, degassed, and baked at 70°C for 4 hours. The PDMS mould was then cut and peeled from the master, punched with a $1.5\,mm$ biopsy punch (Kai Medical) to create inlet ports for tubing insertion. A total of three holes were punched; two inlets for the continuous and aqueous phase and an outlet for waste collection. The PDMS mould was plasma bonded to thin cover slips ($22 \times 50$ mm, $0.13 - 0.17$ mm thick). Hydrophobic surface treatment was performed immediately after bonding by flushing with 1% (v/v) Trichloro (1 H, 1 H, 2 H, 2H-perfluorooctyl) silane (Aldrich) in HFE-7500, and placed in a 65°C oven for 30 min.

### Flow-focusing droplet generation

Microfluidic device fabrication was done using classical soft lithography techniques. A total of 4 devices were developed. All devices consisted of a flow-focusing junction for droplet generation (height, 10 µm), and a second layer with a trapping array made up of circular wells (height, 20 µm) (*Figure 1A*). The trap diameters were 40, 60, 120 and 200 µm. The dimensions of the flow-focusing junction varied depending on trap size. We designed a range of dimensions for the flow focusing junctions and matching trap sizes (i.e. circle diameter). Depending on the trapping size, the total number of traps was between 78 (⌀ = 200 µm) and 840 (⌀ = 40 µm).

Droplets were typically generated at rates approx ~ 50 s⁻¹. The flow rates were controlled using syringe pumps (Nemesys, Cetoni), 1 ML plastic syringes (BD PlastipakTM; sterile needles, 25G x 1" – NR. 18, 0.5 × 25 mm, BD MicrolanceTM 3), and portex tubing PE (Scientific Laboratory Supplies, 0.38 × 0.355 mm). The flow rates for oil and cell suspension were varied depending the size dimensions of each device. A 1:3 ratio was aimed for the continuous and aqueous phase, respectively.

The carrier oil phase was prepared using fluorinated oil HFE-7500 (Fluorochem Ltd) containing 0.25% (w/v) 008-Fluorosurfactant (RAN Biotechnologies, Inc). The aqueous phase consisted of liquid cell cultures (see Cell culturing and maintenance). The droplets were generated at the flow-focusing junction creating water-in-oil emulsions. A dilute suspension of algae was injected through the inlet. Following trapping, droplets were stably confined to the microwell during imaging acquisition. Once all the traps were filled, the aqueous phase flow was halted and the continuous phase flow ratewas reduced approximately 5-fold to flush away excess droplets.

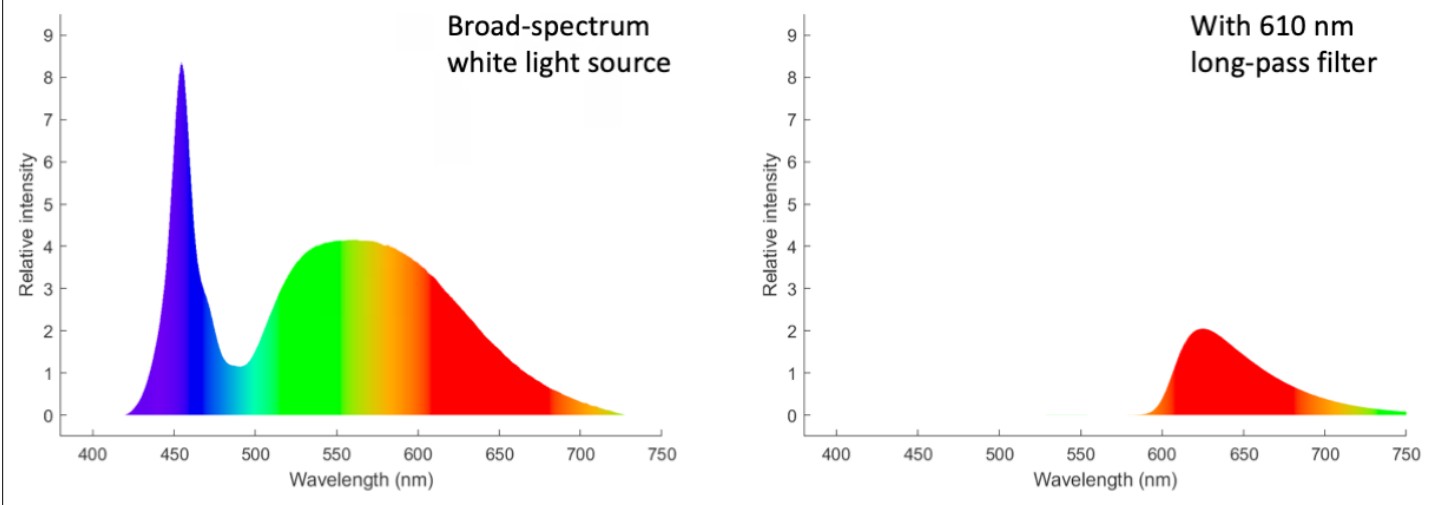

**Figure 8.** Illumination spectra recorded with a spectrometer (OceanInsight OceanHDX, 200 µm-fibre).

## Live-cell high-speed imaging

Brightfield imaging was conducted with an inverted microscope (Leica Microsystems, DMi8), with a high-speed camera (Phantom Vision Research, V1212). We first scanned the array of trapped cells to locate traps matching our criteria (droplet fitting exactly into the trap, droplet containing only one cell). For the 40 µm, 60 µm and 120 µm trap sizes, as well as the cell fusion experiments, we used a 20 x long-working distance objective (HC PL/0.40). For the largest 200 µm traps, we lowered the magnification to 5 x (NPLAN/0.12) equipped with a 1.6 x tube lens, to reduce file size. All traps were imaged with the same intensity and aperture settings, and at 500 fps. For the 1 hour confinement experiments, cells were imaged continuously but 5-min recordings taken at 5 min intervals, to obtain a total of 6 timepoints per cell. Data from droplets that were disrupted at any point during imaging was discarded.

## Light-modulation experiments

For brightfield imaging in WL, we used a standard broad-spectrum LED source to illuminate the specimen. Red light (RL) imaging was accomplished by insertion of an IR long-pass filter (610 nm, Chroma) to the light path. Spectra corresponding to the two possible illumination options are compared in *Figure 8*. For light-switching experiments, the red filter was removed or inserted manually.

## Bulk cell motility assay

We assayed the effect of KCl on *P. octopus* behaviour via a simple open-air method. We first added 1 µL of a concentrated suspension of cells to a glass coverslip under red light in a dark room. We waited $5\,\mathrm{s}$ for flows from the placement of the droplet to subside, before imaging for $25\,\mathrm{s}$ (at 10 x, 100 fps). We then added either a 1 µL droplet of culture media (for the control) or a 1 µL drop of $50\,\mathrm{mM}$ KCl (for the KCl test), and after waiting another $5\,\mathrm{s}$ for flows to subside we imaged for another $25\,\mathrm{s}$.

## Paired-droplet fusion assay

Droplet pairs were generated using a cross junction microfluidic device in which cells and KCl solutions flowing in separate channels are encapsulated in alternation. We used a 25 degrees taper angle previously reported to produce the most stable alternation function (*Saqib et al., 2018*). When stable alternation was achieved, droplets were suddenly halted by removal of the inlet tubings of KCl and cell solutions followed by gentle removal of excess droplets by flowing oil at 3 µL/min for ~1 min. We subsequently identified droplet pairs of expected volumes with one containing a single cell. To ensure fusion with KCl and the absence of mixing prior to fusion, the $10\,\mathrm{mM}$ KCl solution was spiked with 1 µM fluorescein (Merck) which was imaged before triggering of fusion. A 15-min video of the cells was acquired displaying the entrapped droplets 7.5 min prior and 7.5 min post fusion. Fusion was induced by surfactant replacement with 1 H,1H,2H,2H-Perfluoro-1-octanol (Merck) (PFO). A solution of 40%

PFO in HFE was flown at 5 μL/min and run until fusion was achieved (~ 4 minutes). PFO competes with the other surfactant which destabilizes the droplet interface to induce rapid, reproducible fusion. The change in ion concentrations in the solution containing the cell pre- and post-fusion can be found in (*Table 1*). Control experiments where the KCl solution substituted with normal culture medium were performed to confirm the absence of confounding factors.

For the fusion experiments, the identification process consisted of two steps. Firstly we located a trap that had two equal droplets in place, one containing a cell. Secondly, we took a fluorescence image to verify that the droplet without the cell contained KCl. The fluorescence image was taken in the LASX software, using a broad-spectrum LED source (CoolLED-pE300) equipped with a triple-band filter set (including FITC, Ex: 475 nm, Em: 530 nm). The fluorescence intensity was set to 60%, the exposure time was 600 ms, and the gain was 2.0. The presence of fluorescence covering the whole of the empty droplet was sufficient to prove the presence of KCl.

## Image processing and cell tracking

Raw video data was exported to 8-bit grayscale and enhanced by subtracting an average image in MATLAB (Mathworks). Trap boundaries were identified manually to increase the fidelity of 2D cell tracking, which was performed automatically using the TrackMate plugin in ImageJ (*Tinevez et al., 2017*). A Laplace of Gaussian detector was used for spot identification, with slightly different blob diameters for CR and for PO (14 μm and 21 μm respectively). Single continuous tracks were obtained for each experimental run ($N = 5$ individuals per condition), and exported for further processing and extraction of detailed track features/other statistics (see Appendix 1). Video frames from the bulk motility assays were processed and analysed similarly.

## Acknowledgements

This work received funding from the European Research Council (ERC) under the European Union's Horizon 2020 research and innovation programme (grant agreement No 853560 *EvoMotion*, to KYW), and a Springboard Award from the Academy of Medical Sciences and Global Challenges Research Fund (SBF003\1160 to KYW). The research was also funded by the Biological and Biotechnological Research Council (grant BB/T011777/1 to FG). Calculations were performed using the Sulis Tier 2 HPC Platform funded by EPSRC Grant EP/T022108/1 and the HPC Midlands +consortium. We gratefully acknowledge the use of the Lovelace HPC service at Loughborough University.

## Additional information

### Funding

| Funder | Grant reference number | Author |
|---|---|---|
| European Research Council | 853560 | Kirsty Y Wan |
| Academy of Medical Sciences | SBF003\1160 | Kirsty Y Wan |
| Biotechnology and Biological Sciences Research Council | BB/T011777/1 | Fabrice Gielen |

The funders had no role in study design, data collection and interpretation, or the decision to submit the work for publication.

### Author contributions

Samuel A Bentley, Hannah Laeverenz-Schlogelhofer, Resources, Data curation, Software, Formal analysis, Validation, Investigation, Visualization, Methodology, Writing – review and editing; Vasileios Anagnostidis, Resources, Validation, Investigation, Visualization, Methodology, Writing – review and editing; Jan Cammann, Marco G Mazza, Software, Formal analysis, Investigation, Writing – review and editing; Fabrice Gielen, Resources, Supervision, Funding acquisition, Validation, Methodology,

Project administration, Writing – review and editing; Kirsty Y Wan, Conceptualization, Formal analysis, Supervision, Funding acquisition, Validation, Investigation, Visualization, Methodology, Writing – original draft, Project administration, Writing – review and editing

### Author ORCIDs
Samuel A Bentley http://orcid.org/0000-0001-8315-2349
Hannah Laeverenz-Schlogelhofer http://orcid.org/0000-0003-2958-5711
Jan Cammann http://orcid.org/0000-0003-3245-8078
Marco G Mazza http://orcid.org/0000-0002-5625-9121
Fabrice Gielen http://orcid.org/0000-0003-0604-7224
Kirsty Y Wan http://orcid.org/0000-0002-0291-328X

### Decision letter and Author response
Decision letter https://doi.org/10.7554/eLife.76519.sa1
Author response https://doi.org/10.7554/eLife.76519.sa2

---

## Additional files

### Supplementary files
• MDAR checklist

### Data availability
New data and analysis codes generated as part of this study are available for download from Zenodo. The dataset includes all raw cell trajectories and motility states, as well as analysis and simulation codes.

The following dataset was generated:

| Author(s) | Year | Dataset title | Dataset URL | Database and Identifier |
|---|---|---|---|---|
| Bentley SA, Laeverenz-Schlogelhofer H, Anagnostidis V, Cammann J, Mazza MG, Gielen F, Wan KY | 2022 | Dataset for: Phenotyping single-cell motility in microfluidic confinement | https://doi.org/10.5281/zenodo.7226288 | Zenodo, 10.5281/zenodo.7226288 |

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

## Appendix 1

### Analysis of trajectories

Trajectories obtained from cell tracking (see Materials and methods) consist of 2D coordinates at discrete times ($t_1$, $t_2$, $t_3$...) with a constant time interval, i.e. $\Delta t = t_{i+1} - t_i = 0.002s$ for videos captured at 500 fps. The trajectories were analysed using custom MATLAB scripts. First, a smoothing filter was applied. For CR, a Savitzky–Golay filter with order 2 and frame length 201 was used to smooth out the helical trajectories such that a speed and angular velocity corresponding to the net forward motion of the cell could be obtained. For PO, a Savitzky–Golay filter with order 2 and frame length 21 was used. A smaller frame length was used due to the fast timescale of the shock behaviour and since PO has smoother trajectories than CR (*Appendix 1—figure 1*). The cell velocity at time $t_i$ was calculated as

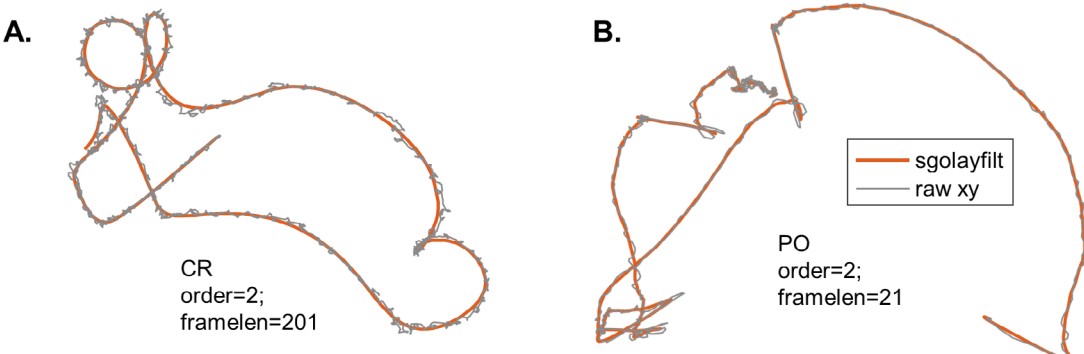

**Appendix 1—figure 1.** Smoothing tracking data using a Savitzky-Golay filter (sgolayfilt). Example raw and smoothed trajectories for CR (**A**) and PO (**B**).

$$\mathbf{v}\left(t_i\right) = \frac{\mathbf{x}\left(t_{i+1}\right) - \mathbf{x}\left(t_i\right)}{\Delta t}, \tag{1}$$

where $\mathbf{x}\left(t_i\right)$ is the 2D coordinate of the centroid of the cell at time $t_i$ for the smoothed trajectories. The angular velocity of the cell was defined as

$$\Omega\left(t_i\right) = \frac{\arccos \hat{\mathbf{v}}\left(t_{i-1}\right) \cdot \hat{\mathbf{v}}\left(t_i\right)}{\Delta t}, \tag{2}$$

where $\hat{\mathbf{v}}\left(t_i\right)$ is the normalised velocity vector at time $t_i$. To reduce the noise of the angular velocity data, the results reported are a moving mean across 25 frames.

Violin plots, which combine box plot and histogram data representations into one diagram, were created using a MATLAB package (*Bechtold et al., 2021*). The width of the violin plots correspond to the probability density function and are scaled linearly for CR and logarithmically for PO.

### Radial probability density

The radial probability density $P\left(r\right)$ is defined as

$$P\left(r\right) = \frac{h(r)/\left(2\pi r \Delta r\right)}{\int_0^{r_{trap}} \frac{h(r)}{2\pi r \Delta r} dr}, \tag{3}$$

where $r$ is the cell's distance from the centre of the trap and $h\left(r\right)$ is the count of the number of trajectory points that lie in a circular shell at distance $r$ with width $\Delta r$.

### Mean square displacement

The the mean square displacement (MSD) is defined as

$$\mathrm{MSD}\left(\tau\right) = \frac{1}{N_\tau} \sum_{t_i} |\mathbf{x}\left(t_i + \tau\right) - \mathbf{x}\left(t_i\right)|^2, \tag{4}$$

where $\tau$ is the delay and $N_\tau$ the number of pairs of time points in the trajectory with $\Delta t = \tau$. The MSD was calculated for the raw 2D coordinate positions using the '@msdanalyzer' MATLAB package (*Tarantino et al., 2014*). The MSD results reported here were calculated using 1/50th of the data points (i.e. every 50th frame) due to the data array size constraints of the package.

The actual trap diameters of the droplets were variable because the resolution of the film mask used to fabricate the PDMS devices was ± 4 µm. The relative error becomes more noticeable for smaller trap sizes and explains the discrepancy between the experimental and simulation results (*Figure 3H*).

## Rotational diffusion

The effective rotational diffusion coefficient $D_R$ was obtained using the mean square angular displacement (MSAD) for CR in the largest trap size (i.e. Ø = 200 µm). First an angle defining the orientation of the cell relative to the y-axis is computed as

$$\theta(t) = \cos^{-1}(\vec{p}(t) \cdot [0, 1]),\qquad(5)$$

where $\vec{p}(t)$ is a unit vector defining the orientation of the cell at time $t$ and, assuming that the cell always swims forwards, is given by $\vec{p}(t) = \vec{v}(t)/|\vec{v}(t)|$, where $\vec{v}$ is the velocity vector. The MSAD is then defined as

$$\text{MSAD}(\tau) = \frac{1}{N_\tau}\sum_{t_i}|\theta(t_i + \tau) - \theta(t_i)|^2,\qquad(6)$$

where $\tau$ is the delay and $N_\tau$ the number of pairs of time points in the trajectory with $\Delta t = \tau$. The data is then fitted to $\text{MSAD} = 2D_R\tau$ for $\tau \le 0.1\text{s}$ in order to obtain an estimate for the effective rotational diffusion coefficient $D_R = 0.93 \pm 0.03\text{s}^{-1}$ (*Appendix 1—figure 2*).

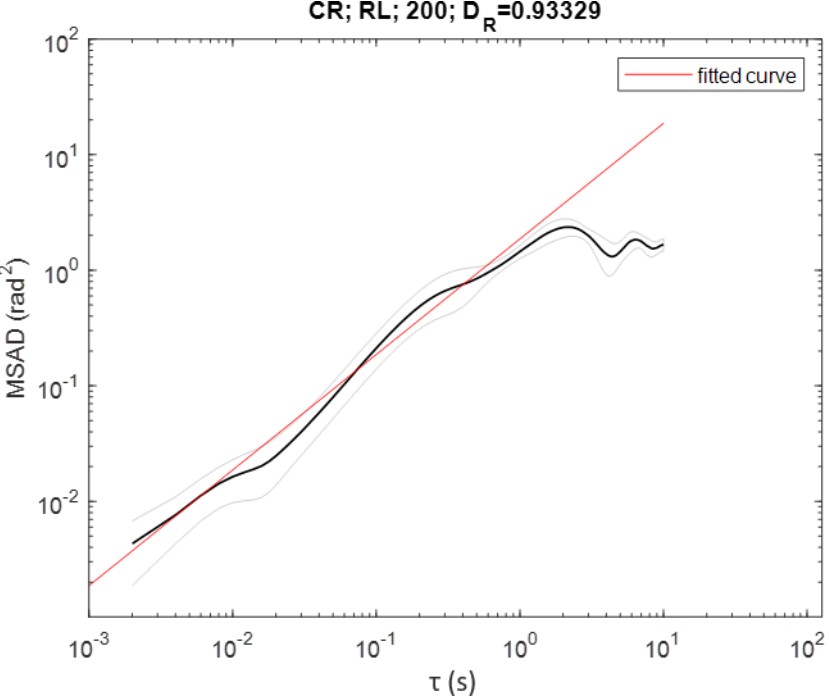

**Appendix 1—figure 2.** The mean square angular displacement (MSAD) for CR in the 200 µm diameter droplets, fitted to $\text{MSAD} = 2D_R\tau$ for $\tau \le 0.1\text{s}$.

## Probability flux calculation and relative probability density

We calculated the probability fluxes using a method previously applied to trajectories of *C. reinhardtii* (*Cammann et al., 2021*) and first introduced by *Battle et al., 2016*. The 2D positional space is divided into a grid of equally sized square boxes with side length $\Delta x = r_{trap}/7$. For each time point the position is assigned to a box $(i,j)$, where $i$ and $j$ are the box positions in the $x$ and $y$

direction respectively. From these coarse-grained trajectories, a time-series of transitions is obtained by constructing the following array

$$
A = \begin{bmatrix}
(i,j)_1 & (i,j)_2 & t_{1,2} \\
(i,j)_2 & (i,j)_3 & t_{2,3} \\
... & ... & ... \\
(i,j)_{N-1} & (i,j)_N & t_{N-1,N}
\end{bmatrix},
\tag{7}
$$

where $(i,j)_n$ and $(i,j)_{n+1}$ are the positions of consecutively visited boxes and $t_{n,n+1}$ is the length of time spent in the initial state $(i,j)_n$ before transitioning to the new state $(i,j)_{n+1}$. In a small number of cases, the two success states $(i,j)_n$ and $(i,j)_{n+1}$ do not correspond to nearest neighbours. In such cases, the intermediate boxes are determined by linear interpolation and the corresponding extra transitions are inserted into the array $A$ (**Equation 7**) to ensure that all transitions in $A$ are between nearest neighbours.

The net transition rates between neighbouring boxes are calculated from the coarse-grained trajectories using

$$
\omega_{(i,j)(k,l)} = \frac{1}{t_{total}} \left( N_{(i,j)(k,l)} - N_{(k,l)(i,j)} \right),
\tag{8}
$$

where $N_{(i,j)(k,l)}$ is the number of transitions from box $(i,j)$ to box $(k,l)$ and $t_{total}$ is the total duration of the trajectory. The net transition rates are then used to calculate the probability flux

$$
j_{(i,j)} = \frac{1}{2\Delta x} \begin{pmatrix}
\omega_{(i,j)(i+1,j)} + \omega_{(i-1,j)(i,j)} \\
\omega_{(i,j)(i,j+1)} + \omega_{(i,j-1)(i,j)}
\end{pmatrix}.
\tag{9}
$$

This method is summarised in **Appendix 1—figure 3**. Note that this calculation does not account for diagonal transitions, however since they account for only < 5% of all nearest neighbour transitions, we assume they have a minimal effect on the results.

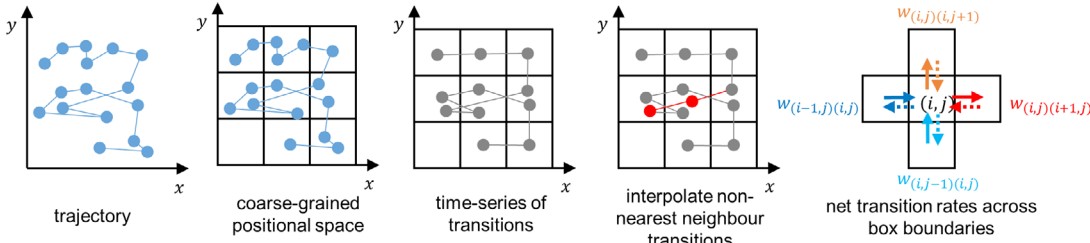

**Appendix 1—figure 3.** Probability flux analysis method.

To obtain the mean tangential flux plotted in **Figures 3I and 4G**, the tangential component of the flux vector was obtained for each box $(i,j)$ and the mean calculated for all boxes with the same radial distance from the centre of the trap.

The 2D trajectory data was also used to calculate the relative probability density

$$
c_{(i,j)} = \frac{A_{trap} n_{(i,j)}}{A_{box} \sum n_{(i,j)}},
\tag{10}
$$

where $n_{(i,j)}$ is the number of trajectory points within box $(i,j)$, $A_{box} = \Delta x^2$ is the box area and $A_{trap} = \pi r_{trap}^2$ is the area of the trap (**Ostapenko et al., 2018**).

## Conversion into three motility macrostates

We started from raw track data, and assigned states in CR and PO using a combination of linear $v$ and/or angular $\Omega$ speeds. In both cases, we took moving averages to reduce frame-to-frame noise (due to the high imaging frame rates).

States for *Chlamydomonas reinhardtii*: (run, stop, tumble)

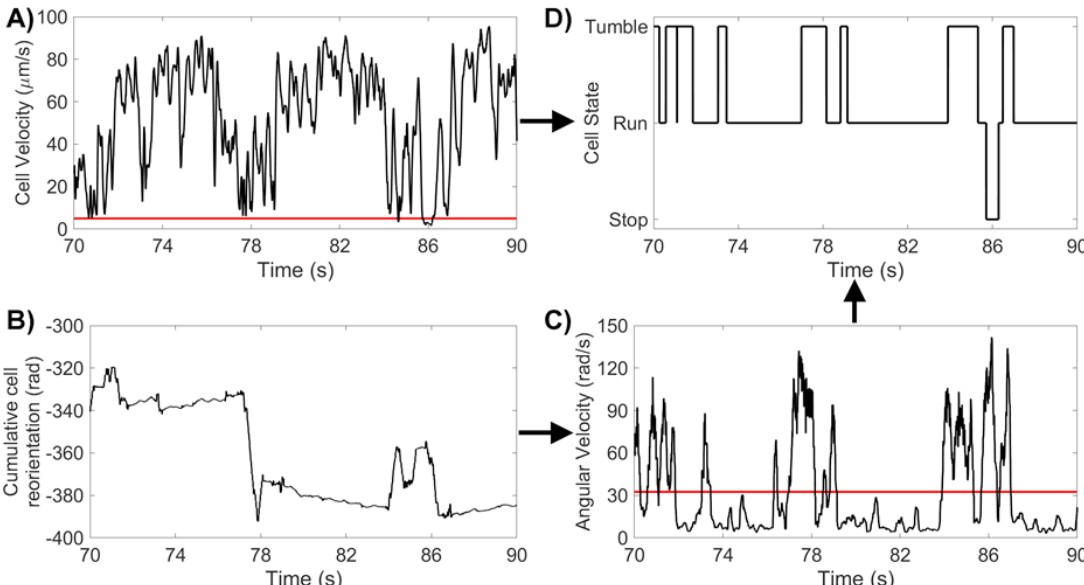

**Appendix 1—figure 4.** Method For Identifying CR States (**A**) CR cell velocity for a 20 s segment, with a run/stop threshold of 5 µms⁻¹. (**B**) CR cumulative reorientation for the same 20 s segment. (**C**) CR angular velocity for the same 20 s segment, with a tumble threshold of $32.5\,\mathrm{rad\,s^{-1}}$. (**D**) CR states for the same 20 s segment.

We defined the 'stop' state as times when $v < 5$ µms⁻¹. A smoothing filter was applied to the binary stop/move data to remove spurious state transitions. We then verified the stop states by visual inspection. To identify 'tumbles', we use angular velocity to locate times when $\Omega > 32.5\,\mathrm{rad\,s^{-1}}$. We filtered the dataset to correct for false tumbles by removing tumbles that were under $200\,\mathrm{ms}$, and for false runs of very short duration found between successive tumbles. We again verified that tumble states corresponded with cell reorientation by visual inspection. Finally, frames that were neither a 'stop' nor a 'tumble', were designated 'runs', with 'stops' taking precedence over 'tumbles'. The workflow is summarised in *Appendix 1—figure 4*.

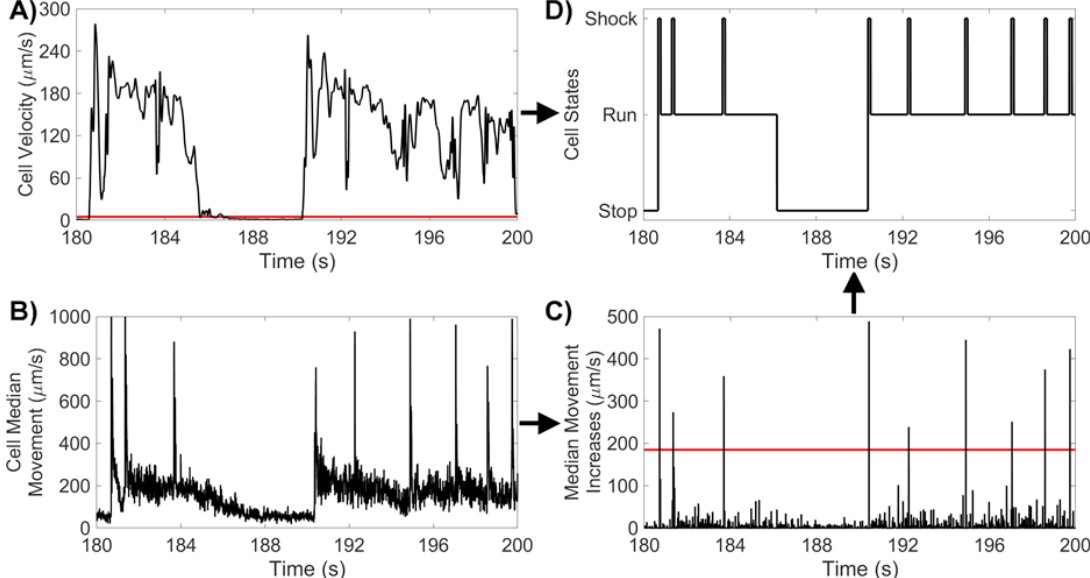

**Appendix 1—figure 5.** Method For Identifying PO States. (**A**) PO cell velocity for a 20 s segment, with a run/stop threshold of 5 µms⁻¹. (**B**) PO median movement over 9-frame windows for the same 20 s segment. (**C**) PO increases in cell median movement for the same 20 s segment, with a shock threshold of 185 µms⁻¹. (**D**) PO states for the same 20 s segment.

## States for *Pyramimonas octopus*: (run, stop, shock)

Here, only the linear speed is sufficient to assign states to PO, since they are associated with very distinct speeds (*Wan and Goldstein, 2018*). We defined the 'stop' state as times when the smoothed speed $v < 5$ µms$^{-1}$. A smoothing filter was applied to the binary stop/move data to remove spurious state transitions. We then verified the stop states by visual inspection.

To identify shocks, we first smoothed the data by computing a local median value and then a moving mean. We then identified each local minimum and the subsequent local maximum, and computed the increase between the two. Increases in median speed greater than 185 µms$^{-1}$ were identified as shocks. The start point of shocks were chosen as the point in time between the local minimum and maximum where half of the total increase in displacement had occurred. Equivalently, the end of the shock was defined as the point in time between the current displacement maximum and post-shock minimum where half of the total decrease in displacement had occurred. Finally, frames that were neither a 'stop' or a 'shock', were designated 'runs', with 'stops' taking precedence over 'shocks'. Brief ($<0.2\,$s) 'run' states between a 'stop' and a 'shock' were reclassified as 'stop' states to remove spurious state transitions. The workflow is summarised in *Appendix 1—figure 5*.

## State probability and transition rate analysis

We estimated transition probabilities between motility macrostates via a simple counting algorithm. State probability is given by

$$p_i = \frac{n_i}{\sum_j n_j}, \tag{11}$$

where $n_i$ is the number of frames in which the cell is classified to be in state $i$.

Transition probability from state $i$ to $j$ is given by

$$p_{ij} = \frac{n_{ij}}{\sum_{k \neq i} n_{ik}}, \tag{12}$$

where $n_{ij}$ is the number of transitions from state $i$ to state $j$.

Transition rate from state $i$ to $j$ is defined as

$$q_{ij} = \frac{p_{ij}}{\langle T_i \rangle}, \tag{13}$$

where $\langle T_i \rangle$ is the mean duration of state $i$.

The survival probabilities for state $i$ are defined by

$$P\left(T_i > \tau\right) = p_i \frac{N(T_i > \tau)}{N(T_i > 0)}, \tag{14}$$

where $N\left(T_i > \tau\right)$ is the number of instances where the duration of state $i$ is longer than $\tau$.

## Beat frequency during a run

The cilia beat frequency for each run period longer than $1\,$s was estimated using a fast Fourier transform analysis of the raw speed (i.e. calculated using the raw centroid positions). A second order Savitzky–Golay filter with frame length 45 was used to reduce the noise of the Fourier transform and the beat frequency for each run was taken to be the highest peak within 30-90 Hz (or if the highest peak was twice the frequency of the second highest peak, then the latter was taken as the beat frequency).

## Numerical simulation of a microswimmer in confinement

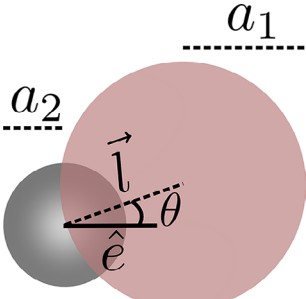

**Appendix 1—figure 6.** In simulations the cell is modelled as an asymmetric dumbbell of two spheres a distance $\vec{l}$ apart. One sphere represents the cell body and the other models the stroke averaged shape of the flagella. To account for differences in the beating patterns of the two flagella the direction of motion is offset from $\vec{l}$ by an angle $\theta$.

The CR cells are modelled as asymmetric dumbbells with a large sphere in front and a smaller sphere in the back, which capture in a minimal way the fore-aft asymmetry of body and appendages (**Roberts and Deacon, 2002**; **Roberts, 2006**, see **Appendix 1—figure 6**). The dumbbell's spheres have different radii $a_{1/2}$ and are separated by a distance $\vec{l}$. Dumbbell models have been used previously to model such microswimmers (**Ledesma-Aguilar and Yeomans, 2013**; **Wensink et al., 2014**; **Wysocki et al., 2015**; **Ostapenko et al., 2018**; **Cammann et al., 2021**). We improve the model by taking into account the slightly different beating patterns of the cis and trans flagella (**Cortese and Wan, 2021**). The asymmetric beating pattern gives rise to an effective chirality in the direction of motion. We implement this by introducing an angle $\theta$ between the swimming direction $\hat{e}$ and the axis of symmetry $\vec{l}$ of the dumbbell. The equation of motion for the position $\vec{r}$ of the active dumbbell is given by

$$\frac{d\vec{r}}{dt} = v_0\hat{e} + \mu_w\vec{F}_w + \vec{\eta}. \tag{15}$$

Here, $v_0 = 80$ and $100\,\mu\text{ms}^{-1}$ is the self-propulsion speed of the cell in red and white light, respectively, and $\vec{\eta}$ is a Gaussian white noise with correlator $\langle\vec{\eta}(t)\vec{\eta}(t')\rangle = 2k_\text{B}T\mu_w\mathbf{1}\delta(t - t')$ with translational diffusion coefficient $k_\text{B}T\mu_w = 250\,\mu\text{m}^2\text{s}^{-1}$ (both values are taken from **Ostapenko et al., 2018**). The term $\vec{F}_w = \vec{F}_1 + \vec{F}_2$ accounts for steric wall interactions, where $\vec{F}_\alpha = -\nabla U_\alpha(r)$, $\alpha = 1, 2$ labels the large and small sphere of the dumbbell, respectively. To compute the steric forces we use the following potential

$$U_\alpha(d)/(k_\text{B}T) = 4\left[\left(\frac{a_\alpha}{d}\right)^2 - \left(\frac{a_\alpha}{d}\right)\right] + 1, \tag{16}$$

if $d < 2a_\alpha$, and 0 otherwise, where $d$ is the distance of the sphere to the wall. The radii of the dumbbell's spheres are $a_1 = 5\mu\text{m}$, $a_2 = 2.5\mu\text{m}$ (see **Ostapenko et al., 2018**). The choice of a softer potential in **Equation 16** is dictated by the rather broad peak in radial probability density compared to the results in **Ostapenko et al., 2018**. The direction of motion $\hat{e}$ is defined with respect to the vector $\vec{l}$ pointing from the small to the large sphere as $\hat{e} = R_\theta\frac{\vec{l}}{|\vec{l}|}$ where $R_\theta = \begin{bmatrix} \cos\theta & -\sin\theta \\ \sin\theta & \cos\theta \end{bmatrix}$. We find that values of $\theta = 1.5°$ (RL) and $-1°$ (WL) model the behavior observed in the different lighting conditions well. The equation of motion for $\hat{e}$ reads

$$\frac{d\hat{e}}{dt} = (\vec{T}_w/\tau_w + \vec{\xi}) \times \hat{e}. \tag{17}$$

Here $\vec{\xi}$ is a Gaussian white noise with correlator $\langle\vec{\xi}(t)\vec{\xi}(t')\rangle = 2D_R\mathbf{1}\delta(t - t')$ and rotational diffusion coefficient $D_R = 0.93\text{s}^{-1}$ (see **Appendix 1—figure 2**). The torque acting at the wall $\vec{T}_w = \vec{T}_1 + \vec{T}_2$, where $\vec{T}_1 = (\vec{r}_1 - \vec{r}) \times \vec{F}_1 = (\vec{l} \times \vec{F}_1)/2$, $\vec{T}_2 = -(\vec{l} \times \vec{F}_2)/2$, and $|\vec{l}| = 5\mu\text{m}$. For the shear time at the wall we use $\frac{\tau_w}{k_\text{B}T} = 0.1\text{s}$ (see **Kantsler et al., 2013**).

