## [Editor Report]

This paper reports on the development of an impressive microfluidic platform for the study of motility, and motility transitions, exhibited by single algal cells in circular confinement. Building on previous work that showed a three-state motility repertoire for certain green algae, the present work uses extremely long time series and a variety of physical perturbations to show how those dynamics can be altered by environmental conditions. The work will be of interest to a wide range of scientists studying motility and non-equilibrium dynamics.

---

## [Decision Letter]

**Decision letter after peer review:**

Thank you for submitting your article "Phenotyping single-cell motility in microfluidic confinement" for consideration by *eLife*. Your article has been reviewed by 2 peer reviewers, one of whom is a member of our Board of Reviewing Editors, and the evaluation has been overseen by Anna Akhmanova as the Senior Editor. The reviewers have opted to remain anonymous. We regret the lengthy delay in furnishing this decision letter.

The reviewers have discussed their reviews with one another, and the Reviewing Editor has drafted this to help you prepare a revised submission. As you will see from the comments below, the reviewers are in agreement about the impressive data sets that have been obtained and the overall methodology, but they both find that a major weakness is the lack of physical/biological insight in the paper in its present form, suggesting that a major revision is necessary.

Essential revisions:

1) The main suggestion to the authors is to take at least some of their detailed observations and analyze them with appropriate theory (as in the confinement problem) to see if agreement can be found (or perhaps to do the theory themselves). Otherwise, in its present form, the paper does not really reach significant conclusions, but instead only holds out the promise of those in the future through the voluminous data that has been acquired.

2) Secondarily, the abstract and indeed a fair amount of the paper is not written in a form that will be easily understood by biologists with only limited background in dynamical systems ideas and the kind of technical jargon that has been used [e.g. words like 'microhabitat' and 'deterministic fusion].

---

## [Author Response]

Essential revisions:1) The main suggestion to the authors is to take at least some of their detailed observations and analyze them with appropriate theory (as in the confinement problem) to see if agreement can be found (or perhaps to do the theory themselves). Otherwise, in its present form, the paper does not really reach significant conclusions, but instead only holds out the promise of those in the future through the voluminous data that has been acquired.2) Secondarily, the abstract and indeed a fair amount of the paper is not written in a form that will be easily understood by biologists with only limited background in dynamical systems ideas and the kind of technical jargon that has been used [e.g. words like 'microhabitat' and 'deterministic fusion].

We thank the editor and reviewers for this summary. In the revision:

1) We have developed a minimal computational model of single-cell movement in confinement, using an approach based on a key study published recently (Cammann et al., PNAS, 2021). This allows for a direct comparison with the previous work. CR-like cells are modelled as fore-aft asymmetric dumbbells with a larger sphere for the flagella and a small one for the body, these then follow overdamped Langevin equations inside a prescribed potential to mimic the presence of the boundary.

In the original submission, we had indicated that our experimental results did not agree with the conclusions of the Cammann et al. paper – regarding the appearance of steady-state flux loops in *circular* confinement. Indeed, while such a swimmer should be expected to perform circular movement, it should do so with equal probability clockwise versus counter clockwise. Therefore, there should be no flux loops averaged over long-times. Here we successfully resolve this discrepancy by introducing a small internal offset, so that the dumbbell becomes left-right asymmetric (in addition to the existing fore-aft symmetry). This is supported by new high-magnification observations of flagellar beating which showed bilaterally asymmetric beating.

This simple adjustment sufficed to reproduce our experimental observations. Including measurements of radial pdfs, msd curves, and also the observed increase in directed fluxes with decreasing trap diameter (as quantified by tangential fluxes). Importantly, it also explains the observed sensitivity of the sense of circulation (whether clockwise or counter clockwise) on light – again implicating that the chirality is under cellular control**,** and therefore related to the cell’s photoresponse. All this is consistent with the fact that CR actively regulates both the sign and magnitude of the asymmetry (flagellar dominance) between its two flagella in order to achieve phototaxis.

We further ruled out any asymmetries in the microwells themselves by performing confocal imaging of trapped droplets, which confirmed that the droplets have a rounded spherical geometry, as illustrated in figure 1B.

These new results have been added to the sections “effect of physical confinement” and “effect of white light stimulation”, figure 3E-I, 4D-G, and all details of the model formulation have been added to the Appendix. Note that the circular plots in figures 2,3,4 have been flipped relative to the original submission to ensure that the viewpoint is always from above the sample in the labframe. Some figure panels from the previous version have been moved to the supplement.

2) We have rewritten the abstract and large parts of the text. We agree that the previous version assumed familiarity with too many concepts from a physics literature. In the revision we removed most references to phrases like ‘deterministic fusion’, ‘microhabitat’, ‘reaction networks’ etc. Where necessary, we have also added explanations to define certain words or concepts when they first appear, before discussing the relevant results.